# Separating the control of moving and holding in human post-stroke arm paresis

Alkis M Hadjiosif[1,2,3]*, Kahori Kita[1,4], Scott T Albert[5,6], Robert A Scheidt[7], Reza Shadmehr[5], John W Krakauer[1,8,9,10]

[1]Department of Neurology, Johns Hopkins University, Baltimore, United States; [2]John A. Paulson School of Engineering and Applied Sciences, Harvard University, Cambridge, United States; [3]Center for Neurotechnology and Neurorecovery, Massachusetts General Hospital, Boston, United States; [4]Arms + Hands Lab, Shirley Ryan AbilityLab, Chicago, United States; [5]Department of Biomedical Engineering, Johns Hopkins University, Baltimore, United States; [6]University of North Carolina School of Medicine, Chapel Hill, United States; [7]Department of Biomedical Engineering, Marquette University, Milwaukee, United States; [8]Department of Neuroscience, Johns Hopkins University, Baltimore, United States; [9]Department of Physical Medicine and Rehabilitation, Johns Hopkins University, Baltimore, United States; [10]Santa Fe Institute, Santa Fe, United States

*For correspondence: ahadjiosif@mgh.harvard.edu

Competing interest: The authors declare that no competing interests exist.

## eLife Assessment

This **important** study extends the previous interesting work of this group to address the potentially different control of movement and posture. Through experiments in which stroke participants used a robotic manipulandum, the authors provide **solid** evidence supporting a lack of a relation between the resting force postural bias they measure (closely related to the flexor synergy in stroke) and kinematic deficits during movement. Based on these results, the authors propose a conceptual framework that differentially weights the two main descending pathways (corticospinal tract and reticulospinal tract) for neurologically intact and stroke patients.

**Abstract** Moving and holding-still (*holding*) have been posited to be separately controlled. The paretic arm after stroke exhibits different abnormalities during rest vs. movement, providing an opportunity to ask whether control of these behaviors is independently affected in stroke. We quantified resting postural abnormalities in human stroke patients by measuring their biases in force production as they held their hand still in various locations in a planar workspace and then assessed whether these resting force biases influenced reaching in the same workspace. Patients displayed marked resting force biases at each location, even when the arm was supported. However, these biases did not transfer to arm-supported planar reaching movements; rather, abnormal resting forces only appeared to switch on after a movement had fully stopped. These findings suggest that moving and holding are functionally separable modes of control. At the same time, resting biases mirrored characteristics of abnormal movement synergies, in line with a shared mechanism. This appears to contradict the functional separation of moving and holding observed in the same patients. To resolve this paradox, we propose a conceptual model that predicts a breakdown in this functional separation when patients move without weight support. This conceptual model posits that synergies are the manifestation of a spillover of posture into movement. Mapping these functional systems onto anatomical and physiological details of lesioned substrate after stroke may provide implementation-level insight into how normal arm motor control is assembled.

## Introduction

A longstanding idea in motor control is that moving and holding-still (abbreviated to *holding* throughout this paper) rely upon separate control regimes. Such separation has been shown in the oculomotor system by the classic work of *Robinson, 1970* who demonstrated that while the controllers for moving and holding are neurally distinct, the hold controller integrates the motor commands that are generated by the move controller. It has been suggested that a separation between these two modes of control may extend to other effectors (*Shadmehr, 2017*). In particular, substantial behavioral (*Ghez et al., 2007*; *Scheidt et al., 2011*) and physiological (*Kurtzer et al., 2005*; *Shalit et al., 2012*) evidence supports the idea that a similar dissociation may govern the control of reaching and holding for the arm (*Jayasinghe et al., 2022*). Moreover, behavioral evidence suggests that commands that are generated during reaching are integrated to produce a component of the commands that are generated during subsequent holding (*Albert et al., 2020*). This has raised the possibility that, like the control architecture of the oculomotor system, a holding controller may lie downstream of the reaching controller.

Patients with hemiparesis after stroke typically exhibit dissociable deficits in the control of reaching and holding within and across effectors (*Garland et al., 2009*; *Levin, 1996*; *Mani et al., 2013*; *Schaefer et al., 2009*; *Scheidt and Stoeckmann, 2007*; *Trombly, 1992*; *Zackowski et al., 2004*), making hemiparesis a potent model for understanding the interaction, or dissociation, between these

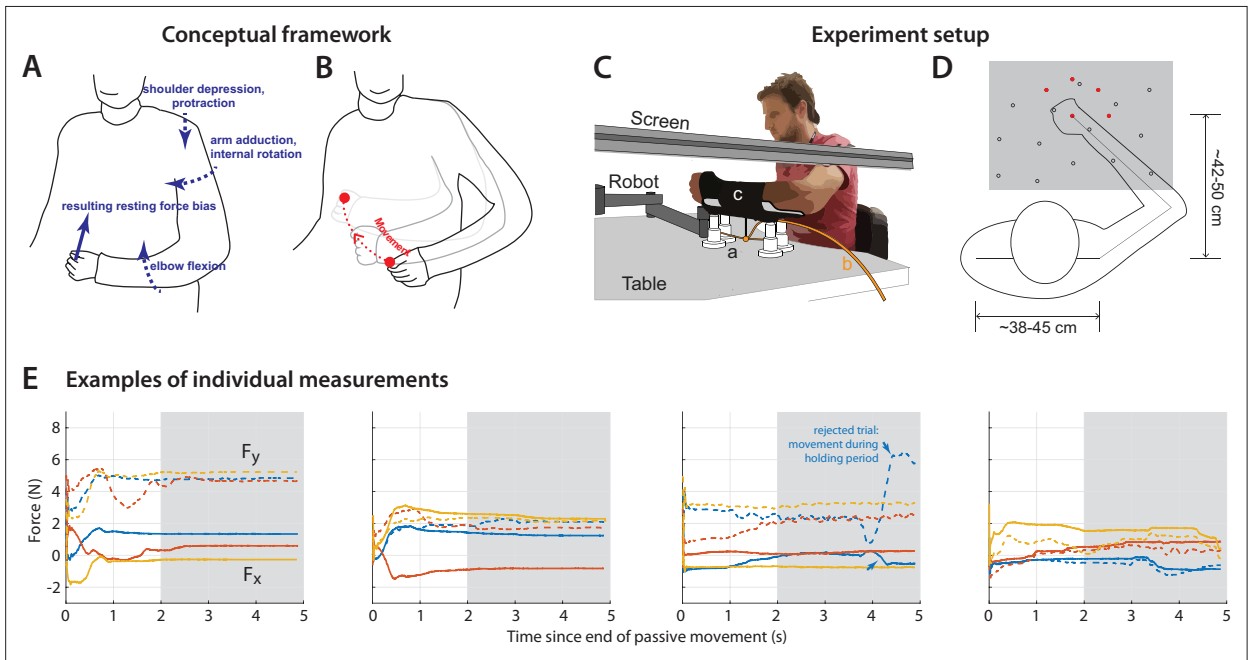

**Figure 1.** Framework and experiment setup. (**A**) A patient exhibiting a typical flexor posture at rest. Dashed arrows indicate elements of the posture: shoulder depression, arm adduction/internal rotation, elbow flexion. If one were to physically constrain the hand in a position away from the resting posture, the torques involved in each component of the abnormal resting posture translate to a force on the hand (sketched as a blue arrow); we thus designed an experiment to measure the resting force bias on the hand, as a marker of the overall postural abnormality. The goal was to compare resting postural force biases to active movement control in the same area (**B**). (**C**) Experiment setup. The participant holds the handle of the robotic arm; reach targets and cursor position are projected on a screen on top; for arm support, the participant's arm is strapped on an armrest (**c**) connected to an air sled (**a**) which rests on the table. Air is provided through the tube labeled (**b**). (**D**) Top-down view of setup, illustrating the different hand positions where resting postural forces were measured in Experiment 1 (open circles). Also shown are the five target positions used in the reaching and holding task for Experiment 2 (filled red circles). The gray box indicates the workspace depicted in *Figure 2*. (**E**) Examples of measurements of resting force biases. Each panel shows the evolution of resting forces during the 5 s holding period for one participant (same participants as in *Figure 2*), taken at positions close to midline and distant from the body, under the same condition (paretic arm, arm support given). Solid lines indicate the force on the x-axis (positive values indicate forces towards the left), whereas dashed lines indicate the force on the y-axis (positive values indicate forces towards the body). The shaded area indicates the time window over which forces were averaged to estimate the resting bias, illustrating how resting biases were relatively stable by the 2 s mark. Note that the third panel includes one trial (blue) which was rejected following visual inspection as described in Materials and methods – Data Exclusion Criteria, due to instability in force production and movement during the hold period.

two modes of motor control. The magnitude of reaching and holding abnormalities can be relatively large, making it easier to measure and compare them.

Here, we focused on one particular aspect of holding: resting at a position. One of the most common and readily recognizable motor symptoms after stroke is abnormal resting posture (*Twitchell, 1951*). For example, the typical hemiparetic arm posture consists of flexion at the fingers, wrist, and elbow (*Carr and Kenney, 1992*; *Figure 1A*). Prominent approaches for the treatment of hemiparesis have been based on the idea that abnormal resting posture has a direct deleterious effect upon movement control; they advocate for adjusting overall posture in ways that may minimize such effects (*Bobath, 1982*; *Carr and Kenney, 1994*; *Carr and Kenney, 1992*).

Our main aim was to determine whether resting abnormalities bleed over into active movements in the post-stroke arm. This is of great interest because the ability to separate moving from holding may be precisely what is lost after stroke, with postural abnormalities contaminating voluntary movement. If this is indeed the case, it might lend credence to the idea in rehabilitation that treating resting abnormalities can benefit voluntary movement.

In Experiment 1, we assessed resting postural abnormalities by measuring resting postural force biases in patients with stroke using a planar workspace. We investigated how these force biases varied with arm position, presence of arm support, and overall motor impairment. In Experiment 2, we proceeded to assess patients' control of reaching and holding in the same workspace. We separately investigated effects on initial reach and bringing the reach to a stop, as these two might be separately controlled (*Ghez et al., 2007*; *Scheidt and Ghez, 2007*). We also investigated active holding control *after* the movement was over, by examining responses to perturbations that attempted to move the arm off the target, in order to confirm that the same controller is engaged for both passive (as in Experiment 1) and active holding at the same position.

## Results

### Patients exhibited resting force biases across the whole workspace

We first assessed post-stroke resting postural abnormalities across a 2D workspace. In Experiment 1, participants grasped the handle of a robotic arm, which passively moved their hand to a series of positions that sampled the workspace in front of them (the setup and example workspace are shown in *Figure 1C and D*). Participants were instructed to maintain grasp, but otherwise relax their arm and not resist the actions of the robotic arm. Once the participant's arm was passively moved to a given position, the robotic arm would hold still for 5 s, enabling us to measure the forces involuntarily exerted by the participant as their hand was held at that location. Each experimental block consisted of three visits to 15–25 positions sampled for each participant, in random order. Participants completed four different experimental blocks: two with each arm, with or without arm weight support (provided by an air sled, *Figure 1C*).

Patients displayed abnormal postural force biases when the robot held the hand still at various locations across the workspace. *Figure 2* shows examples of this for three patients and a typical healthy control participant. The non-paretic arm produced little to no postural force biases, whereas the paretic arm produced substantial postural force biases, particularly when the hand was held in a more distant position. The postural force biases were strongest when patients had to support the weight of the arm against gravity. Moreover, the patient with the highest degree of impairment (top left subplot of *Figure 2*, as assessed using the Fugl-Meyer score for the Upper Extremity, FM-UE *Fugl-Meyer et al., 1975*) exhibited the strongest resting postural abnormalities.

To aggregate results across all participants, we focused on abnormal resting postural force biases at five specific hand locations, illustrated by the red dots in *Figure 2*. These locations were chosen as they were within the sampled workspace of all participants (as this workspace could differ from one participant to the next due to different participants' segment lengths), represented positions both near and distant to the body, and, importantly, they were also the movement targets used in Experiment 2. *Figure 3A* shows subject-averaged resting postural forces at each of these five positions, and *Figure 3B* shows the corresponding force magnitudes and directions averaged across subjects and the five positions. There were two key observations: first, postural force biases were stronger at locations further away from the body (average magnitude – paretic side – without arm support: 6.5±1.3 N for the three more distant targets vs. 3.5±0.5 N for the two more near targets, $t_{15}=3.5$,

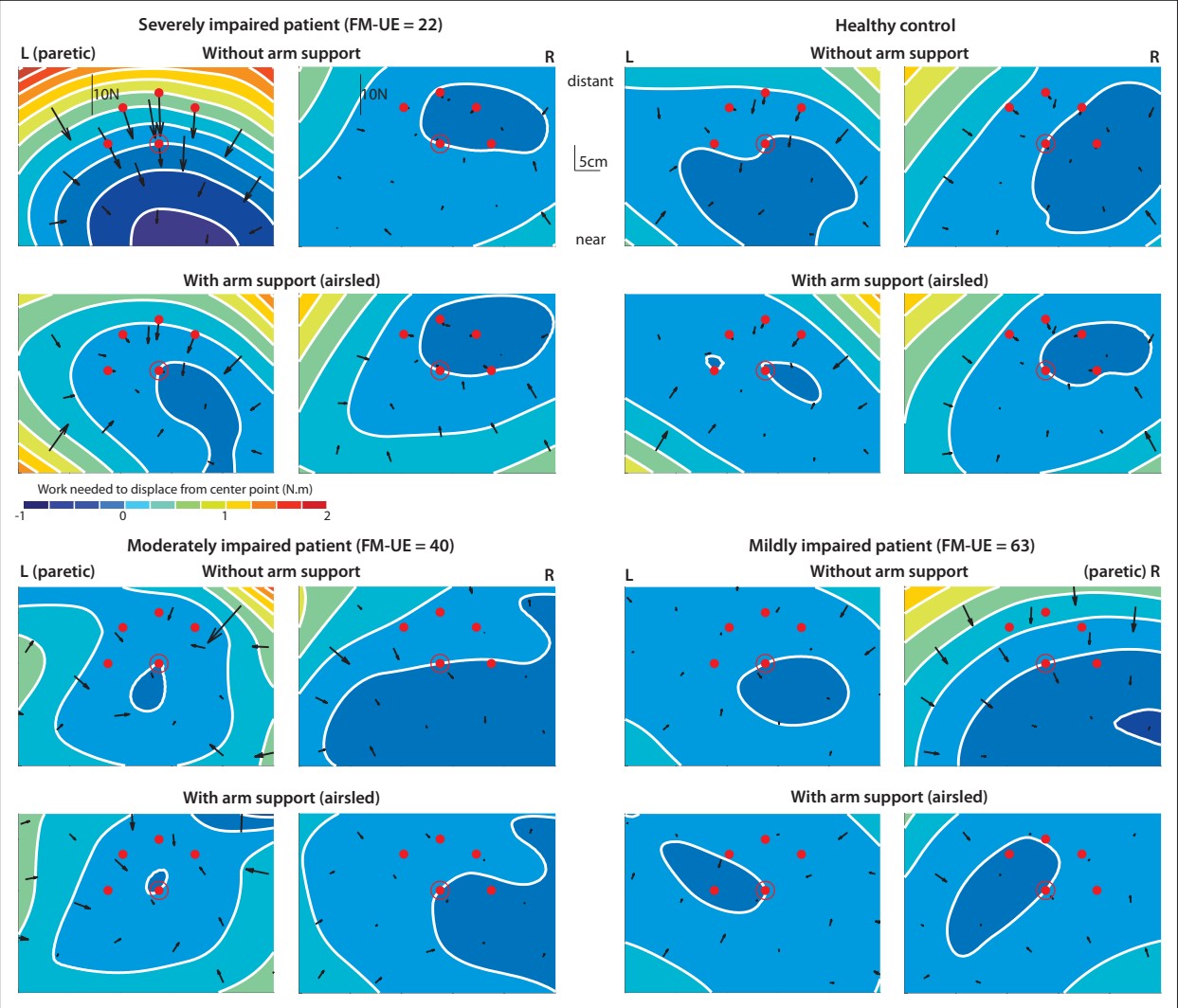

**Figure 2.** Examples of resting postural force biases. Shown are three stroke patients and one healthy control. Arrows indicate magnitude and direction of abnormal resting postural forces as measured at the hand at each location. Isoclines and the corresponding color levels provide a visual representation of where these biases would tend to direct the hand (from red towards blue). These isoclines represent different levels of the (spatial) integral of the posture bias-field (with zero referring to the isocline passing through the center position). Postural bias force vectors cross these isoclines perpendicularly. Please see the shared analysis code at OSF for details on how this visual aid was constructed. The red dots are the reach targets, with the center location circled (used in Experiment 2). Note how abnormalities in the paretic side are considerably stronger when arm support is removed. FM-UE: Fugl-Meyer score for the Upper Extremity (0–66).

p=0.0030; with arm support: 3.3±0.5 N for the three more distant targets vs. 1.9±0.3 N for the two more near targets, $t_{15}$=3.7, p=0.0022) and tended to point towards it (average direction – paretic side: −93.3±6.0° unsupported vs. −136.5±21.6° supported, with −90° directed towards the body); second, postural force biases were roughly halved in magnitude when arm support was provided (average magnitude – paretic side: 5.3±0.9 N unsupported vs. 2.8±0.4 N supported, $t_{15}$=4.0, p=0.0013).

## Examining the influence of resting postural force biases on active motor control

In Experiment 2, we sought to investigate whether resting postural force biases influence active reaching and holding in the same workspace. Specifically, we measured their effects upon initial reach direction (*Figure 4*, blue) and approach direction when bringing the reach to a stop (*Figure 4*, red), as it has been suggested that these two components are separately controlled (*Ghez et al., 2007*; *Scheidt and Ghez, 2007*); see also *Karst and Hasan, 1991*; *Lestienne, 1979*. We also asked whether

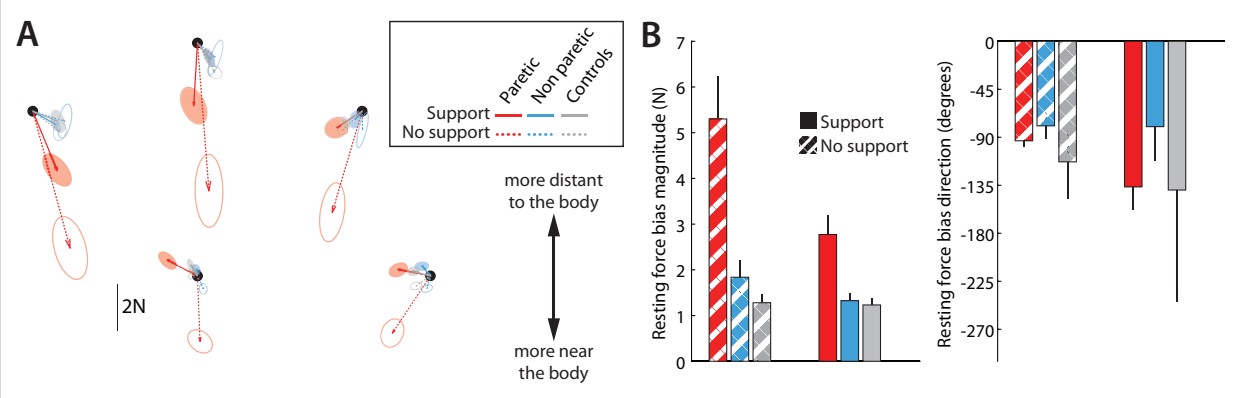

**Figure 3.** Average resting postural force biases. (**A**) Average resting postural forces for the paretic (red) and non-paretic (cyan) arms of patients (n=16), as well as control participants (gray, n=9), illustrating how abnormal forces in the paretic arm are stronger in more distant targets and attenuated when arm support is provided (lighter shades). To average across left- and right-hemiparetic patients, left-arm forces were flipped left to right. (**B**) Corresponding average resting postural force magnitudes and resting postural force directions (0 indicating the 3 o'clock direction, with negative values indicating clockwise directions). Error bars indicate mean ± SEM (circular mean ± SEM in the case of movement directions).

the same controller is engaged while the arm is passively (Experiment 1) vs. actively (Experiment 2) held at the same position. For this purpose, we examined active holding control *after* the movement was over (*Figure 4*, black), using perturbations that attempted to push the arm off the target.

## Resting postural force biases did not seem to affect the control of active reaching in the same workspace

We began by examining the control of active reaching. Participants made 10 cm point-to-point reaching movements within an array of five positions (filled circles in *Figure 1D*), for a total of eight different movement directions (*Figure 5A*). Arm support was provided by the air sled (*Figure 1C*). Patients' movements were generally impaired, taking more time and traveling a longer path to reach the target than controls (Time to target: 1.59±0.12 s for patients' paretic side vs. 0.82±0.03 s for controls, p=0.000063; Path traveled to target 13.4±0.6 cm for patients' paretic side vs. 10.8±0.2 cm for controls, p=0.0071, *Figure 5B and C*).

We then examined whether patients' movements were affected by the resting postural biases measured in Experiment 1 in the same workspace and under the same arm-support condition. Would there, for example, be a difference when moving the arm through a high-postural bias area vs. a low-postural bias area? We examined effects of resting postural biases upon the initial reach vs. the approach to a hold position, as separate mechanisms may be involved in the control of each phase of movement (*Ghez et al., 2007*; *Hannaford and Stark, 1985*; *Karst and Hasan, 1991*; *Sainburg et al., 1999*; *Scheidt and Ghez, 2007*); resting postural biases might affect one phase but not the other. Specifically, we examined: (a) whether the direction of resting postural forces at the start position influenced trajectory deviations near the beginning of the movement (*Figure 5D*) or (b) whether direction of resting postural forces at the end position influenced trajectory deviations near the end of the movement (*Figure 5G*, after the participant reached within 2 cm of the target).

To investigate within-subject effects, we selected, for each participant, the two movement directions for which the corresponding postural forces had strongest opposing effects – that is the most rightwards (CW) vs. most leftwards (CCW) with respect to movement direction. The idea behind picking the most extreme values for each individual was to maximize our sensitivity in detecting potential effects of resting postural forces upon active movement. Our analysis found no significant differences in directional biases between these two conditions: while the selected start-point force biases differed considerably between the most CCW and CW instances (1.9±0.4 N vs. –2.1±0.4 N, correspondingly [negative signs indicating CW forces], $t_{15}$=5.68, p=0.000044), the corresponding initial angular deviations did not (0.4±2.1° for the most CCW vs. –1.6±2.2° for the most CW postural force, $t_{15}$=0.56, p=0.58, *Figure 5D–F*). Similarly, while the selected end-point force biases also differed considerably (2.1±0.4 N vs. –1.9±0.4 N, for the most CCW vs. CW instances, $t_{15}$=5.68, p=0.000044), the corresponding endpoint angular deviations did not (3.6±2.3° for the most CCW vs. 4.9±2.6° for

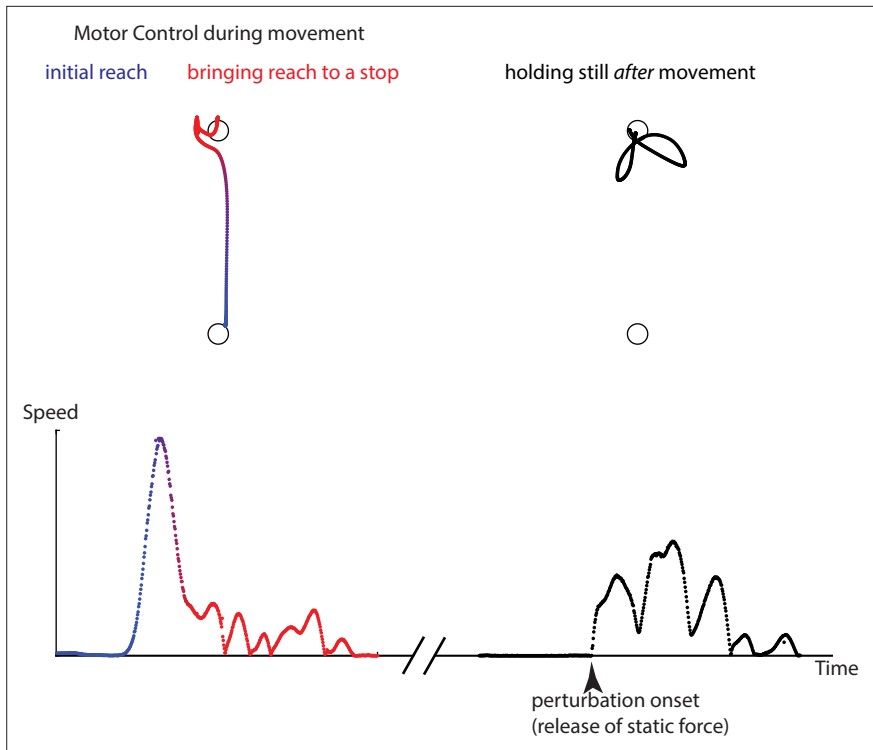

**Figure 4.** Three aspects of active motor control that we tested in Experiment 2. We separately examined the early part of the reaching movement (blue) and the late part, when the arm was coming to a stop (red). This was done by studying both unperturbed movements at different stages and movements that were perturbed with brief force pulses. In addition, we examined active holding control after the movement was over (black), using perturbations that tried to move the arm away from the held point. Shown is an example of trajectory and speed profiles from the reaching and coming-to-a-stop parts of a trial (left) and active holding against a perturbation after the trial was over (right).

the most CW postural force, $t_{15}$=0.50, p=0.62, *Figure 5G–I*). *Figure 5—figure supplement 1* shows the same analyses but for non-paretic and control data, illustrating, like in the paretic data, a lack of evidence that resting posture biases affect active movement.

While selecting the most extreme instances in terms of resting posture biases for each individual provides the greatest bandwidth to detect effects of such biases upon active movement, it also does not use the majority of the data (the 6 out of 8 movement directions corresponding to intermediate strengths of posture biases). We thus conducted an additional analysis which estimated the sensitivity (slope) of initial or endpoint deviations against the corresponding resting biases using the entirety of the data. In line with our main analysis, we found that patients' slopes were not significant in either case (Initial reaching angle vs. posture bias at start: 1.5±1.4°/N, p=0.30; endpoint reaching angle vs. posture bias at endpoint: –1.0±0.8°/N, p=0.25, *Figure 5—figure supplement 2*), further illustrating the lack of compelling evidence that postural abnormalities directly affect active movement.

A potential limitation in our data and analysis above is that unperturbed reaching movements may provide limited sensitivity in detecting effects of resting postural force biases, as any such effects may be largely compensated by a predetermined motor plan (*Scheidt et al., 2011*). To address this, we probed the effect of the biases to responses to less predictable, externally applied mid-movement force perturbations. In 1/3 of reaching movements in Experiment 2, chosen randomly, we imposed a 70ms bell-shaped force pulse which was 12 N at its peak and acted lateral to the movement (*Smith and Shadmehr, 2005*). These pulses were imposed 2 cm into the reach. Half of these pulses were clockwise (CW, blue in *Figure 6A*) and the other half were counter-clockwise (CCW, red in *Figure 6A*). We first verified that these pulses had a clear effect upon movement: perturbed movements took longer to complete in both patients (paretic movement time: 1.72±0.13 s vs. 1.59±0.12 s, $t_{15}$=5.89, p=0.000030) and controls (movement time: 0.88±0.03 s vs. 0.82±0.03 s, $t_8$=5.56, p=0.00054). Patients

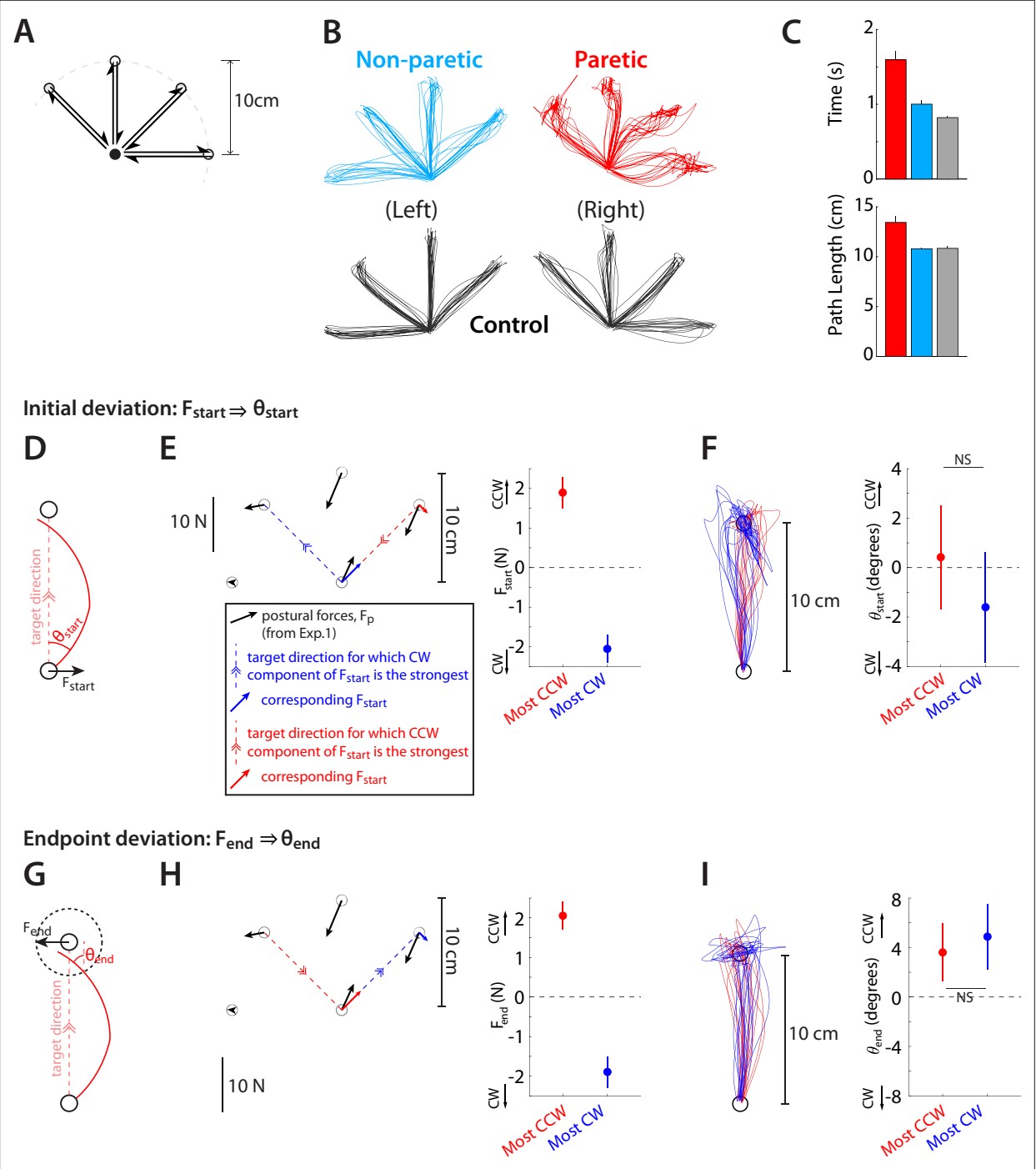

**Figure 5.** Abnormal resting postural force biases do not interact with active reaching. (**A**) Target array for Experiment 2 (movement task), illustrating the 5 start/end points of reaches and the 8 movement directions. (**B**) Example outwards trajectories (unperturbed trials) for a patient (cyan: non-paretic side; red: paretic side) and a healthy control (gray). (**C**) Subject-averaged reach performance based on either time (top) or path length to target (bottom) indicates impaired reaching control in patients' paretic side. (**D–F**) Within-subject analysis of whether resting postural forces at movement start bias early movement towards their direction. (**D**) Sketch illustrating the concept behind this analysis. Assuming a movement from a start position subject to a strong rightwards resting bias ($F_{start}$), will that translate to a corresponding rightwards movement bias which can be expressed as the early reach angle $\theta_{start}$? (**E**) For each individual, we selected the target direction for which the counter-clockwise (CCW) component of $F_{start}$ was the strongest (red) vs. the target direction for which the clockwise (CW) component was the strongest (blue). The left panel shows this selection for an example participant: postural forces at start position were projected lateral to the movement direction, allowing us to select movement directions for which the lateral component was directed the strongest CCW or CW. The right panel shows the magnitude of these selected components across all patients. (**F**) Left: Corresponding movement trajectories (rotated so start position is at the bottom and end position at the top) for the directions selected for

*Figure 5 continued on next page*

*Figure 5 continued*

the same example participant. Right: Average initial angular deviations, $\theta_{start}$, across the selected directions for each participant. Note the lack of difference between the instances for which the CCW vs. the CW component of $F_{start}$ was the strongest, and thus no evidence that $F_{start}$ impinges upon the movement. (G–I) same as D-F but for endpoint resting postural forces, $F_{end}$ and endpoint deviations, $\theta_{end}$. Error bars indicate SEM. Data from n=16 stroke patients and n=9 healthy control participants. Comparisons in F and I indicate paired t-tests; NS=not significant.

The online version of this article includes the following figure supplement(s) for figure 5:

**Figure supplement 1.** Relationships between active reaching and resting posture biases for the non-paretic side of patients and for controls.

**Figure supplement 2.** Estimating the sensitivity of active reaching to resting posture biases.

**Figure supplement 3.** Across-individual correlations between resting bias and movement kinematics.

**Figure supplement 4.** Repetition of the analysis in *Figure 5E/F* (top) and 5 H/I (bottom) but with resting biases calculated without trial rejection, showing similar results (lack of relationship between $\theta_{start}$ and $\theta_{end}$ with $F_{start}$ and $F_{end}$, respectively).

generally had impaired response to these pulses compared to their non-paretic side and healthy controls, deviating further (maximum lateral deviation – paretic: 1.25±0.05 cm vs. non-paretic: 1.03±0.03 cm [p=0.00077] and controls: 1.07±0.04 cm [p=0.0357] - average of CW and CCW pulses) and taking a longer time to stabilize in the pulse direction (i.e. settling time; paretic: 0.61±0.03 s vs. non-paretic: 0.48±0.01 s [p=0.0027] and controls: 0.50±0.01 s [p=0.028]). This is illustrated in *Figure 6B and C*.

We then investigated whether resting postural forces played any role in patients' response to the pulse perturbation. We reasoned that, should resting postural forces play a role, they would tend to decrease the effect of the pulse if they were in the opposite direction to it and increase the effect of the pulse if they were in the same direction. This is because force pulses acting against the gradient of the postural bias field would act to stretch the already active muscles, further increasing postural resistance; whereas force pulses acting along the gradient of the postural bias field would act to shorten the same active muscles, reducing postural resistance. We thus compared the lateral component of resting postural forces in the start position against (a) the maximum lateral deviation in the direction of each pulse and (b) the time taken to stabilize in the pulse direction (settling time). In a manner similar to the previous section, we selected, for each individual, the directions where the starting lateral postural force was most CCW vs. most CW and compared the corresponding deviations. We found no clear effect: when resting postural forces were the most opposed to the pulse (vs. most aligned with it) there were no clear differences in deviation along the pulse direction, for neither the CCW pulses (0.97±0.17 vs 1.30±0.14 cm, $t_{15}$=1.48, p=0.16), the CW pulses (1.27±0.16 vs 1.50±0.20 cm, $t_{15}$=0.87, p=0.40), or with both types of pulses pooled together (1.12±0.12 vs 1.40±0.11 cm for the most-opposed vs. most-aligned pulse/bias instances, $t_{15}$=1.42, p=0.18). Similarly, we found no clear differences in settling time between the instances where resting postural forces were the most opposed to the pulse vs. most aligned with it, for neither the CCW pulses (0.61±0.05 vs 0.64±0.05 s, $t_{15}$=0.40, p=0.70), the CW pulses (0.59±0.07 vs 0.65±0.07 s, $t_{15}$=0.58, p=0.57), or with both types of pulses pooled together (0.60±0.05 s vs. 0.64±0.05, $t_{15}$=0.74, p=0.47). For both metrics, we got similar findings when examining non-paretic and control data (*Figure 6—figure supplement 1*). In line with the above analyses, we also found no effects when we instead estimated the sensitivity of either outcome variable against the corresponding resting biases using data from all trials (for CCW and CW pulses combined, maximum lateral deviation along the pulse direction 0.07±0.06 cm/N, p=0.21; settling time: 0.00±0.02 s/N, p=0.92, *Figure 6—figure supplement 2*).

In summary, we found no evidence for an interaction between resting postural force biases and the ability to respond to perturbations that were applied during movement.

## Resting postural force biases emerged during active holding at the end of movement

Finally, in Experiment 2, we also investigated the relationship between resting postural force biases and active holding. In 20 out of 96 movements in each block, participants had to hold steady on the target for an additional 5–7 s (*Figure 7A*). During this time, the robot imposed a 6 N force in one of four directions (45°, 135°, 225°, 315°, as shown in *Figure 7B*). This force was gradually applied over two seconds, held at a 6 N level for 3–5 s, and then abruptly released, acting to displace the arm in the direction opposite to the original force, as illustrated in the examples in *Figure 7B*.

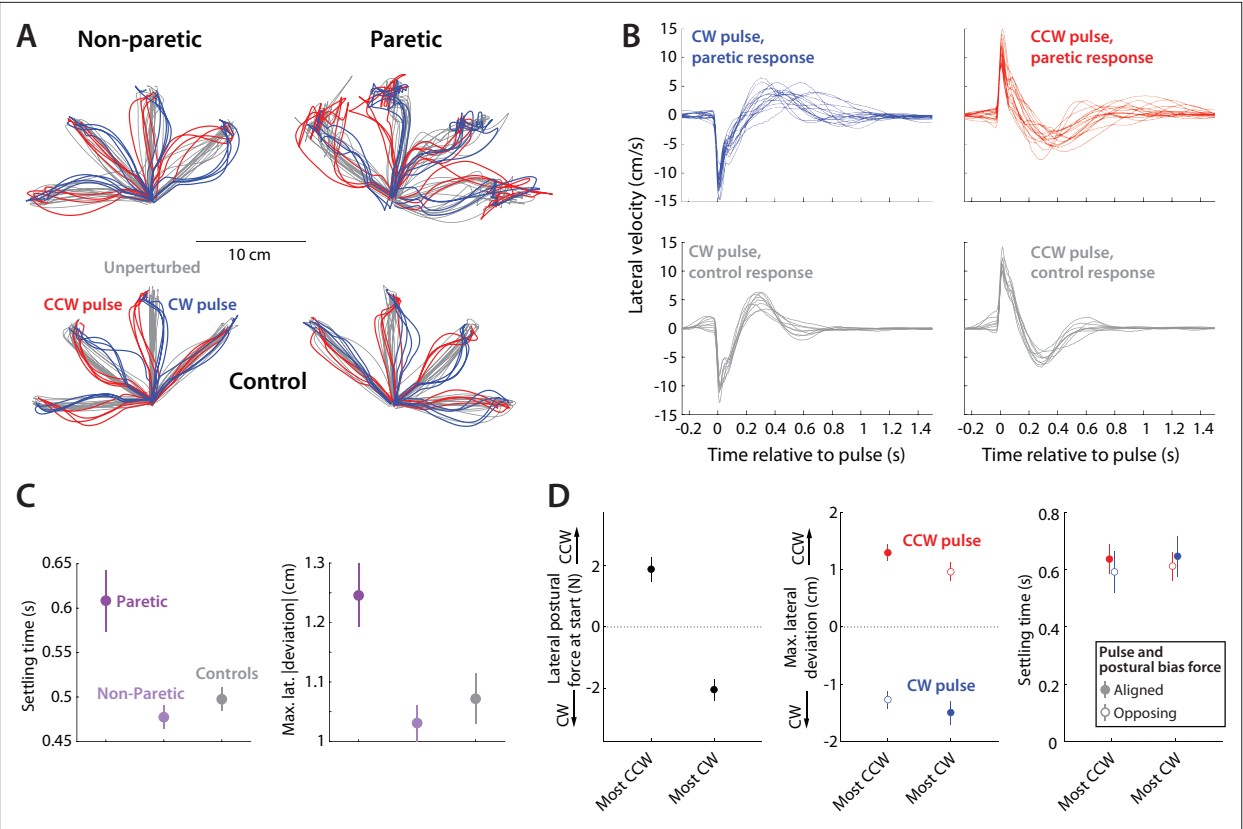

**Figure 6.** Responses to pulse perturbations during movement are not affected by resting postural force biases. (**A**) Examples of perturbed (red: perturbed with CCW pulse; blue: perturbed with CW pulse) and unperturbed (gray) outward trajectories - same individuals as in *Figure 3B*. (**B**) Lateral velocity (positive: CCW to movement) before and after pulse onset, and corresponding responses from controls (gray), illustrating how patients, in response to the pulse, take longer time to settle and tend to experience larger lateral deviations compared to controls. (**C**) Summary performance measures for patients and controls, indicating impaired performance with the paretic side: settling time (left) and maximum lateral deviation on pulse direction (right). (**D**) Within-individual analysis: here, for each individual, we selected the movements for which the starting-position resting postural force would be either the strongest CCW or CW (left); we then examined the corresponding settling time (middle) and maximum lateral deviations (right). We find no effects of the most CCW vs. most CW resting postural forces in either case: there is no evidence for either reduced settling time or increased maximum lateral deviation for instances where pulse and resting bias are most opposing (open circles) compared to the instances where pulse and resting bias are most aligned (filled circles). Error bars indicate SEM; data from n=16 stroke patients and n=9 healthy control participants.

The online version of this article includes the following figure supplement(s) for figure 6:

**Figure supplement 1.** Relationships between responses to pulse perturbations and resting posture biases for the non-paretic side of patients and for controls.

**Figure supplement 2.** Estimating the sensitivity of movement perturbation responses to resting posture biases.

**Figure supplement 3.** Across-individual correlations between resting bias and movement perturbation responses.

**Figure supplement 4.** Repetition of the analysis in *Figure 6D* but with resting biases calculated without trial rejection, showing similar results (lack of effect of resting biases upon the response to the force pulse).

Patients showed impaired capacity to resist and recover from this perturbation (the abrupt release of the imposed force). The time to stabilization for the paretic side (0.94±0.05 s) was longer compared to the non-paretic side (0.79±0.03 s, p=0.024) and controls (0.78±0.06 s, although this was statistically marginal, p=0.061) as shown in *Figure 7E*, left. Moreover, patients traveled a longer path to stabilization (7.7±0.7 cm for the paretic side vs. 5.9±0.3 cm for the non-paretic side [p=0.012] and 5.4±0.5 cm for controls [p=0.026], *Figure 7E*, middle) and they deviated more in the direction of the perturbation (3.3±0.2 cm for the paretic side vs. 2.6±0.1 cm for the non-paretic side [p=0.012] and 2.3±0.2 for controls [p=0.0078], *Figure 7E*, right).

To investigate whether resting postural force biases affected the control of actively holding on different workspace locations, we performed a within-individual analysis analogous to the one we

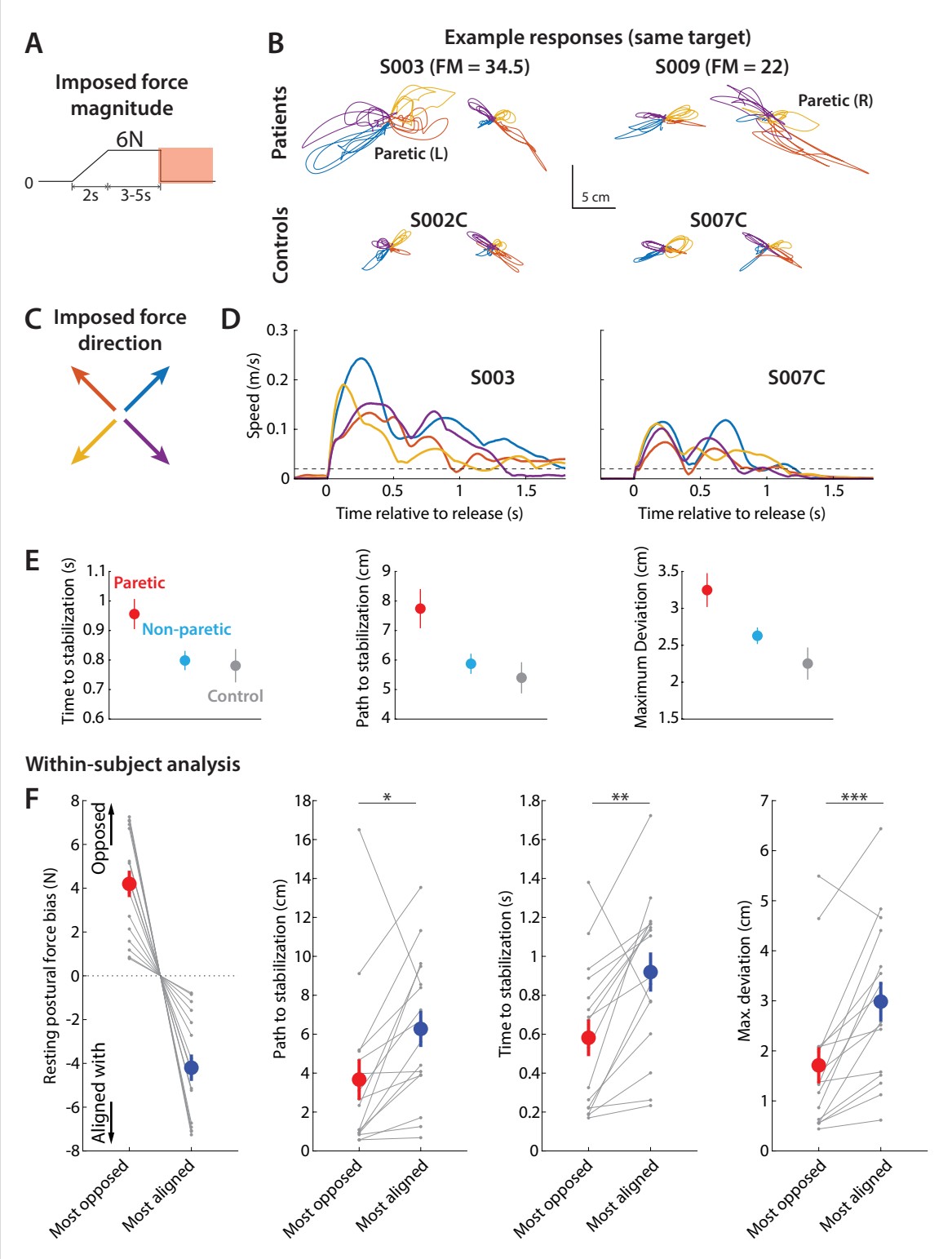

**Figure 7.** Responses to release perturbations during holding and their relationship to resting postural forces. (**A**) Time course of the perturbation. (**B**) Example responses (all for the same position in the workspace) from two patients (top row) and two controls (bottom row). (**C**) Corresponding imposed force directions, the abrupt removal of which perturbs the movement in the opposite direction (compare with B). (**D**) Examples of speed profiles after the sudden release of the imposed hand force, averaged for all trials at the same position for each participant. A dashed line indicates the 2 cm/s threshold used to assess time to stabilize. Left; example patient (paretic side); Right; example control. Colors correspond to different directions of the

*Figure 7 continued on next page*

*Figure 7 continued*

imposed hand force. (**E**) Summary of performance metrics after the perturbation for the paretic and non-paretic side of patients (n=16) and healthy controls (n=9). (**F**) Within-subject analysis of the relationship between resting postural forces in the direction of the perturbation vs. performance against the perturbation. For each individual, we selected the two position/perturbation direction combinations for which resting postural forces were either the most opposed (red) to the perturbation or the most aligned (blue) with it. From left to right: forces in selected position/perturbation direction combinations; corresponding path traveled to stabilization; corresponding time to stabilization; corresponding maximum deviation. Note how the most-opposed resting bias for each patient is equal and opposite to their most-aligned resting bias. This is because the same resting bias, when projected along the direction of two oppositely directed perturbations (illustrated in C), would oppose one with the same magnitude it would align with the other. This analysis suggests that resisting postural perturbations and restoring hand position after the perturbation was indeed easier when resting postural forces opposed, rather than were aligned with, the perturbation. Gray dots indicate individual data; colored dots and error bars indicate mean ± SEM. Comparisons indicate paired t-tests; *p<0.05; **p<0.01; ***p<0.001.

The online version of this article includes the following figure supplement(s) for figure 7:

**Figure supplement 1.** Relationships between responses to static release perturbations and resting posture biases for the non-paretic side of patients and for controls.

**Figure supplement 2.** Estimating the sensitivity of active holding control to resting posture biases.

**Figure supplement 3.** Across-patient comparisons between average postural biases (projected along the direction of the release perturbation) and different metrics of performance against the release perturbation: path to stabilization (left column); time to stabilization (middle column); and maximum deviation (right column).

**Figure supplement 4.** Time course of responses to the static release perturbation.

**Figure supplement 5.** Repeating the analyses in *Figure 7F* to ensure no systematic effects of missing data.

**Figure supplement 6.** Repetition of the analysis in *Figure 7F* but with resting biases calculated without trial rejection, showing similar results (performance against the static release perturbation is better when the resting biases are directed against the perturbation, and worse when the resting biases are aligned with the perturbation, showing interaction between resting biases and active holding control).

used for unperturbed reaches and moving perturbations. We first projected resting postural forces upon the directions of the release perturbation to assess the component of the resting postural force that opposed or aligned with the release perturbation. We then selected, for each participant, the two position/perturbation direction combinations for which these forces were either the most opposed (*Figure 7F*, red) to the perturbation or the most aligned (*Figure 7F*, blue) with it. For these selected position/perturbation direction combinations, we compared capacity to resist and recover from the perturbation and found that this capacity was indeed better when the resting postural force was in a direction that opposed the perturbation (path traveled to stabilization: 3.7±1.0 cm vs. 6.3±0.9 cm, $t_{15}$=2.8, p=0.014; time to stabilization: 0.6±0.1 s vs. 0.9±0.1 s, $t_{15}$=3.8, p=0.0017; maximum deviation: 1.7±0.4 cm vs. 3.0±0.4 cm, $t_{15}$=4.6, p=0.00036, corresponding to instances with the most opposed vs. the most aligned resting postural force). We did not observe significant differences when we performed the same analyses for non-paretic and control data (*Figure 7—figure supplement 1*).

We then proceeded, in a secondary analysis, to estimate the sensitivity of active holding control to resting postural biases for each individual using data from all trials (*Figure 7—figure supplement 2*). Specifically, we estimated the sensitivity of all three outcome variables (path to stabilization, time to stabilization, and maximum deviation) against the resistive component of the resting bias for each trial. Consistent with our main analysis in the previous paragraph, we found significant sensitivities in all cases (path to stabilization: sensitivity of –0.69±0.30 cm/N, p=0.034; time to stabilization: sensitivity of –0.050±0.009 s/N, p=0.000089; maximum deviation: –0.30±0.12 cm/N, p=0.027). The negative slopes indicate reductions in these metrics – that is, better performance – when the corresponding resting bias is more resistive. In short, our perturbations revealed that resting flexor biases 'switched on' after movement was over, providing evidence for separate control between moving and holding.

## Direct comparison of effects of resting postural force biases on moving vs. holding perturbations

The analysis above compared the effects of resting biases on moving vs. holding indirectly, finding effects in one case but not the other. We thus proceeded to directly compare how the two types of perturbations (moving vs. holding) interact with resting biases. An obstacle in this comparison is that the magnitude of perturbation responses may be different for moving vs. holding perturbations. This could be because the magnitude of the perturbations themselves is different, or because of other

factors such as overall stiffness being different between the moving and holding state. To address this, we calculated a Response Asymmetry Index (RAI):

$$RAI = \frac{r_A - r_O}{r_A + r_O}$$

Here, $r_A$ is the response in the direction where resting bias is most-aligned with the perturbation, and $r_O$ is the response in the direction where resting bias is most-opposed to the perturbation.

The idea behind the RAI is that, while the magnitude of responses may differ between the two types of perturbation, this will be accounted for by the ratio used to calculate the asymmetry. Similar approaches have been used to assess symmetry/laterality across a variety of different modalities (*Cramer et al., 1997*; *McPherson et al., 2018*; *Robinson et al., 1987*).

We calculated RAIs for two response metrics used for both types of perturbation: maximum deviation and time to stabilization/settling time. For the maximum deviation (*Figure 8*, left side), there is more asymmetry for the holding case, although the p-value is marginal (p=0.088), likely due to the large variability in the pulse case (individual values shown in black dots). For time to stabilization/settling time (*Figure 8*, right side), the difference is significant (p=0.0048). Together, these analyses indicate that resting biases interact substantially more with holding compared to moving, in line with a relative independence between these two control modalities.

## Relating resting postural force biases to the Fugl-Meyer scale for the upper extremity

The observation, from Experiment 1, that resting postural force biases are reduced by external arm support bears parallels to the same effect arm support has on abnormal synergies active during movement (*Sukal et al., 2007*). Here, 'synergies' refer to abnormal co-activation patterns across joints that manifest as the patient tries to move – for example, the elbow involuntarily flexing as the patient tries to abduct their shoulder (*Brunnstrom, 1966*; *Twitchell, 1951*). Yet, Experiment 2 found no relationship between resting postural force biases and active movement control. To further investigate this apparent paradox, we examined the relationship between resting postural force biases and abnormal synergies in further detail. To assess the level of synergy abnormalities for each patient, we measured their Fugl-Meyer scores for the upper extremity (FM-UE), a scale which was designed to capture abnormal muscle synergy after stroke (*Brunnstrom, 1966*; *Fugl-Meyer et al., 1975*) and closely corresponds to EMG-based synergy measures (*Bourbonnais et al., 1989*).

We entered patients' resting postural bias magnitudes into a mixed-effects ANOVA with FM-UE (continuous), Proximity (distant: the three locations furthest from the body, near: the two locations closest to the body, *Figure 9*, top left) and Support (with air sled, without air sled) as factors, evaluating for main effects and interactions. All factors showed significant main effects, with resting postural force magnitudes decreasing with FM-UE ($\eta^2$=0.27, p=2 × 10$^{-8}$), target proximity ($\eta^2$=0.13, p=0.00003), and weight support ($\eta^2$=0.13, p=0.00004). Significant interactions were observed between FM-UE and both Support (p=0.0062) and Proximity (p=0.0034), with Support and Proximity becoming more important

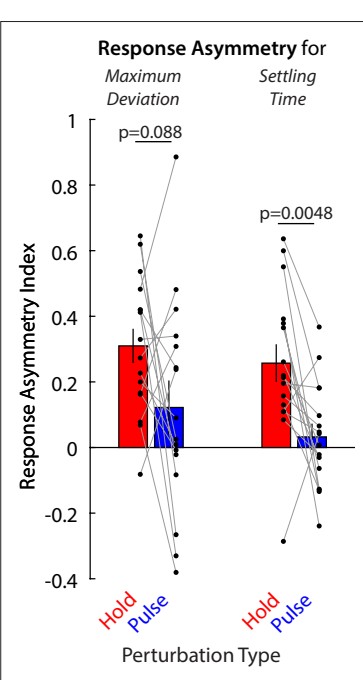

**Figure 8.** Direct comparison of the effects of resting force biases on holding perturbations vs. pulse perturbations. Response Asymmetry Indices (RAIs) are shown for holding (red) vs. moving/pulse (blue) perturbations. Positive values indicate a response asymmetry in line with an effect of resting force biases. Individual data (n=16 patients) are shown in black dots. Error bars indicate SEM. Comparisons indicate paired t-tests.

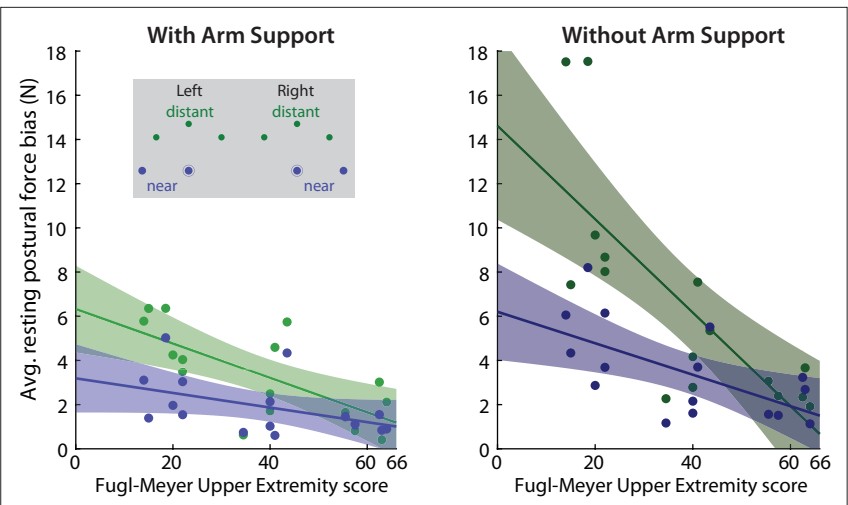

**Figure 9.** Relationship between resting force biases and abnormal synergies. Across-patient (n=16) relationships of FM-UE (/66, higher scores indicating lower impairment) and resting postural force magnitudes, for distant (green) and near (blue) target positions, with (left) and without support (right). Note the strong effects of arm support, proximity, and FM-UE. Lines indicate linear fits; shading indicates 95% confidence interval for each fit.

for lower FM-UE scores (i.e. higher overall motor impairment / higher synergy intrusion) as illustrated in *Figure 9*.

In summary, we made three key observations with regard to abnormal resting postural force biases. First, like abnormal synergies, they were exaggerated when active arm support was required. Second, they were more pronounced in more distant positions where the elbow was more extended; if patients were to actively reach to the same locations, they would have faced increased intrusion of flexor synergy. Third, they scaled with the synergy-based FM-UE. These observations suggest a common mechanism behind resting postural force biases and abnormal synergies.

## Discussion

We assessed abnormal resting posture in stroke patients by measuring the resting force biases they involuntarily exerted while their arm was held at different points within a planar workspace (Experiment 1). We found that these resting postural force biases were strongest in more distant positions of the arm, generally pulled the arm toward a flexed position, and were significantly reduced, but nevertheless remained present, with arm weight support. We then proceeded to assess reaching and holding control in the same workspace in the presence of arm support (Experiment 2) and examined whether resting postural forces could partially account for deficits in the motor control of reaching and holding in the same patients. Remarkably, resting postural force biases did not have a detectable effect on the control of active reaching and only emerged during the control of holding after the movement ended. This suggests a dissociation between the control of movement and posture. At the same time, assessing patients' impairment using the FM-UE, a metric designed to measure for abnormal synergies during 3D arm movements, revealed a strong association between resting force biases and abnormal movement synergies, which raises the possibility that the observed dissociation of movement and posture control for planar weight-supported movements may break down for unsupported 3D arm movements. This dissociation raises interesting questions about both the neural architecture responsible for these two forms of control and how to approach rehabilitation of the post-stroke arm.

### Dissociation between reaching and holding

Previous research provides evidence for separate control of reaching and holding in the healthy arm – for a review, see *Shadmehr, 2017* and (*Jayasinghe et al., 2022*). For example, following visuomotor rotation training specific to the outward phase of an out-and-back movement, participants did not transfer this rotation to holding after a point-to-point reach in the same direction: after the

movement was over, and visual feedback was removed, participants' held position drifted from the rotated movement endpoint towards the baseline hold position (*Scheidt and Ghez, 2007*). There is also neurophysiological evidence for separate control of moving and holding. A previous study found populations of neurons in macaque M1 that represent mechanical loads during posture or movement but not both (*Kurtzer et al., 2005*), whereas in another study cortical neurons and spinal interneurons coded parameters related to either movement or posture maintenance (*Shalit et al., 2012*).

The dissociation reported here between the control of reaching and holding in the post-stroke arm is consistent with a recently proposed hybrid model, which posits distinct controllers for reaching and for bringing the arm to a stop (*Jayasinghe et al., 2022*). Our finding that resting posture control does not interact either with the initial reach or bringing the arm to a stop extends this idea, suggesting *three* distinct controllers: one for the initial reach; another for bringing the arm to a stop; and another for control of holding at the endpoint after movement is over. It should be noted, however, that having distinct neural circuits for reaching and holding does not rule out interactions between them. For example, we recently demonstrated how arm holding control reflects the integration of motor commands driving the preceding active movement that led to the hold position, in both healthy participants and patients with hemiparesis (*Albert et al., 2020*). However, in that paper, we did not claim that this integration is the only source of holding control. Indeed, in Experiment 1 of the current study, we used passive movement to bring the arm to each probed position, which means that the postural biases could not be the result of integration of motor commands.

## Potential origins of abnormal resting flexor biases

Experiment 1 revealed several parallels between resting postural force biases and abnormal synergies: a propensity for flexion, mitigation by arm weight support, and a significant correlation between patients' resting postural force biases and their FM-UE scores. The FM-UE is a measure designed to assess post-stroke abnormal synergies during active movement. These similarities raise the possibility that post-stroke resting postural biases and movement synergies share a generative mechanism. Consistent with this possibility, a study found that externally imposed elbow flexion led to (involuntary) shoulder flexion and external adduction in stroke patients but not healthy controls, suggesting abnormal synergy patterns do not require active voluntary movement to be expressed (*Sangani et al., 2007*) – also see *Schmit and Rymer, 2001*.

It has been proposed that abnormal resting posture after brain injury can be ascribed to an extrapyramidal system (i.e. other than the corticospinal tract, CST; *Denny-Brown, 1964*). Consistent with this suggestion, multiple lines of evidence from animal models point towards reticulospinal tract (RST) involvement in postural control (*Deliagina et al., 2007*; *Lacquaniti et al., 1997*; *Takakusaki, 2017*). Lesions of the monkey ponto-medullary reticular formation resulted in abnormal postures; notably, these were characterized by trunk and limb flexion, elevated shoulders, and arms held close to the body (*Lawrence and Kuypers, 1968*) mirroring the flexor posture pattern seen after stroke. Several studies in cats also suggest a postural role for the reticulospinal tract. Reticular formation neurons respond to vestibular inputs such as head tilts or whole-body tilts, in line with a role in adjusting posture against gravity (*Bolton et al., 1992*; *Matsuyama and Drew, 2000*; *Pompeiano et al., 1984*). Electrical and chemical stimulation of the reticular formation leads to modulations in muscle tone (*Takakusaki et al., 2016*), and reticulospinal neurons display tonic activity patterns that are related to postural adjustments that precede a movement rather than to the movement itself (*Schepens and Drew, 2004*).

There is also recent evidence that the RST is the descending system responsible for the generation of large forces during voluntary movement (*Glover and Baker, 2022*; *Tapia et al., 2022*). Thus, the accumulated evidence suggests that the RST could control posture and large force production in the upper limb. Upregulation of the RST has also been implicated in the generation of abnormal movement synergies after stroke (*McPherson et al., 2018*). For example, the ipsilateral RST can facilitate flexors but suppress extensors, a pattern mirroring the flexor synergy (*Davidson et al., 2007*; *Davidson and Buford, 2004*).

There appears to be a contradiction, however, between the idea that abnormal postures share a common mechanism with abnormal movement synergies, and our finding that there was no evidence for intrusion of resting biases into any phase of active reaching. A potential explanation for this apparent contradiction is that our experiments were conducted on a 2D surface with weight support.

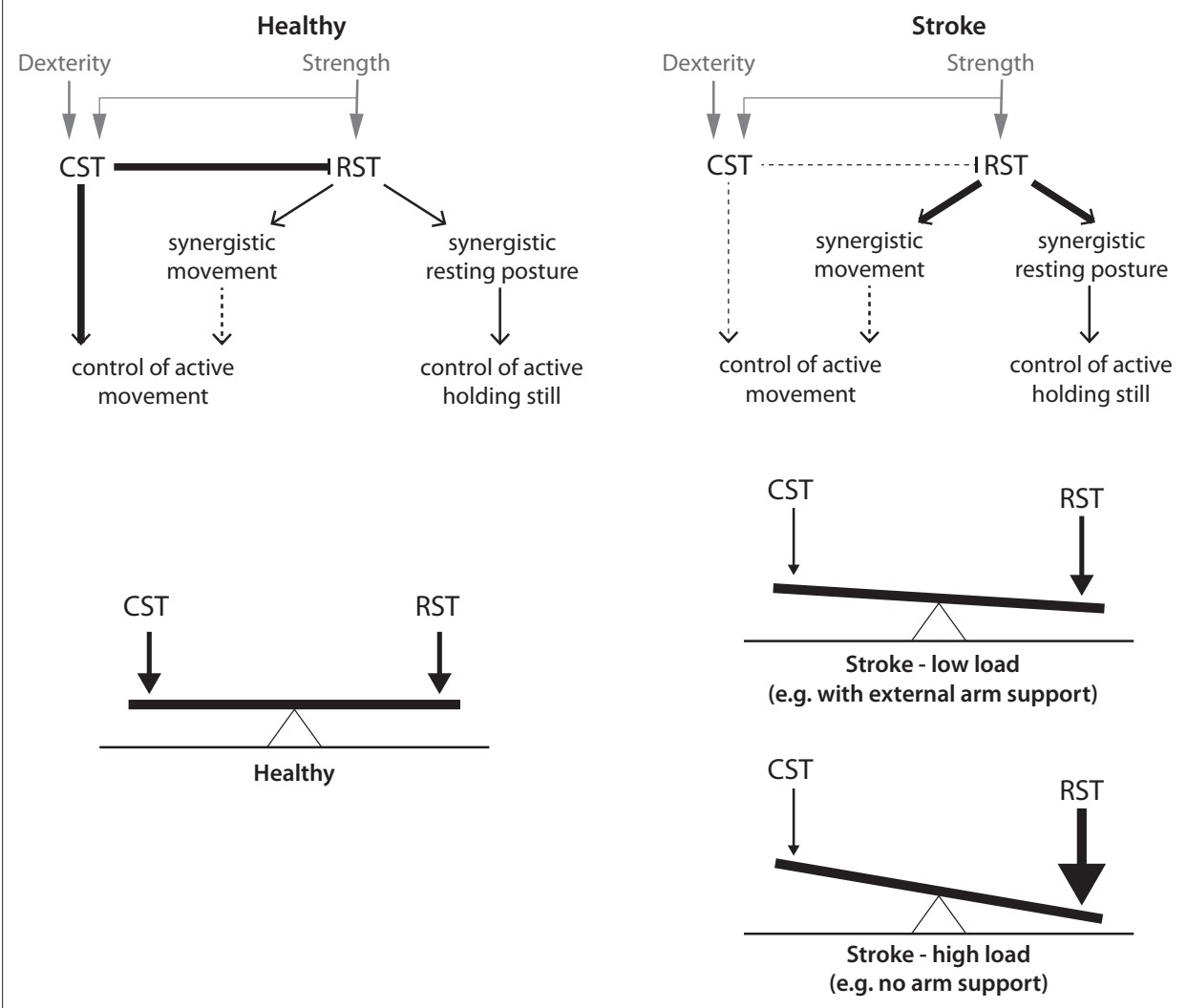

**Figure 10.** An architecture for the separable control of reaching and holding and spillover effects in stroke.

In contrast, the FM-UE scale, which we used as a measure of abnormal synergy, is performed by patients in 3D without weight support. It has been shown that for 3D reaching, patients with chronic stroke express intrusive flexor synergies (*Zackowski et al., 2004*). Weight support – as was used in Experiment 2 – reduces intrusion of flexor synergies for planar movements (*Sukal et al., 2007*) and improves planar kinematics (*Beer et al., 2007*; *Beer et al., 2004*). Earlier work, using a weight-supported planar task, found similar degrees of reaching abnormality whether movements were made in or out of synergy (*Levin, 1996*); more recently, we showed that reaching dexterity can be dissociated from synergy intrusion when arm support is provided (*Hadjiosif et al., 2022*). Thus, while full weight support reduces both resting flexor biases and movement-related flexor synergies, this reduction is more complete for synergies compared to resting biases. This is not inconsistent with positing a shared substrate for abnormalities at rest and during movement, but this substrate might play a greater role in holding than in moving, and so the consequences of damage to it may be more apparent in a task that stresses holding over movement.

What would this framework look like? We posit that the motor system has separable functional modes for moving vs. holding and that this is accomplished by differentially weighting the contributions of descending systems that are operative in both modes (*Figure 10*). The CST is weighted more towards fast and fractionated control during movement, whereas the RST, in contrast, is weighted more towards slower postural control and generation of large isometric forces (such as vertical forces for arm support, or horizontal forces for holding the arm still against a background load like in our

posture/release perturbation trials). In healthy individuals, these two modes are kept in balance by the CST, which has a moderating influence on the RST (*Schepens and Drew, 2006*; *Figure 10*, left). In this framework, the CST is the controller during movement and the modulator during holding.

Post-stroke damage to the CST reduces the moderating influence the CST has upon the RST (*Figure 10*, right); influence that is likely further compromised by upregulation of the RST through plasticity mechanisms (*García-Alías et al., 2015*; *Zaaimi et al., 2012*). This conceptual model can explain our results as follows: As the CST is the dominant system during movement, it can still modulate the RST in this mode, especially with weight support, as this reduces RST drive, with the consequence that resting biases do not markedly contaminate active movement. Conversely, the RST is the dominant system for postural control and can overcome weakened CST modulation, which leads to the resting biases we observed. Thus, weight support allows a weakened CST to keep moving protected from holding, but it cannot prevent abnormal holding itself. The interesting implication of this conceptual model is that synergies are, in fact, postural abnormalities that spill over into active movement when the CST can no longer modulate the increased RST activation that occurs when weight support is removed (i.e. resting biases may influence active reaching in absence of weight support). Supporting this idea, a study found increased ipsilateral activity (which primarily represents activation via the descending ipsilateral RST *Zaaimi et al., 2012*) when the paretic arm had reduced support compared to full support (*McPherson et al., 2018*).

A side question is how the control of decelerating to a stop – which we show here is distinct from the control of holding after the movement is over – fits within such a scheme. A recent saccade study found that deceleration may be controlled through the cerebellum (*Sedaghat-Nejad et al., 2022*). In reaching tasks in the mouse, activity in the interpositus nucleus scaled with limb deceleration (*Becker and Person, 2019*), whereas disruption of the pontine nuclei did not impair movement initiation as much as kinematic variables related to bringing the movement to a stop (*Guo et al., 2021*).

Increased recruitment of the RST has also been implicated in power grip (*Baker and Perez, 2017*; *Tazoe and Perez, 2017*). This may be relevant to our study, as our task had participants actively grasp the handle of the robot for both Experiments 1 and 2. If increased RST recruitment indeed explains abnormal resting postural biases as we discussed in the previous section, a stronger grasp would in turn increase the strength of these biases even further. Moreover, the intermixing of free-reaching and perturbation trials in Experiment 2 could have led to increased uncertainty in environmental dynamics; uncertainty can lead to adjustments such as even stronger grip (*Hadjiosif and Smith, 2015*). Yet, despite this potential additional RST recruitment, we found that resting biases did not affect active reaching and only switched on after the reach was over; detected when we applied a perturbing hand force. This result lends further support for separate controllers for reaching vs. holding.

## Limitations

Another potential source for the resting postural force biases we observe could be abnormally low thresholds (*Levin and Feldman, 1994*; *Powers et al., 1989*) or abnormally high gains (*Thilmann et al., 1991*) of the stretch reflex. Passively extending the elbow, even at low speeds – something Experiment 1 did – can lead to increased muscle activity which may persist after the end of movement (*Kanade-Mehta et al., 2023*; *Levin et al., 2000*). This velocity-dependent increase in muscle tone – spasticity – could potentially explain some of the resting force biases measured in Experiment 1. However, in an earlier task where the elbow was passively extended at different velocities, the resulting tonic biceps EMG terminated around the time that the applied elbow extension was completed (*Thilmann et al., 1991*). And, in recent work, *Kanade-Mehta et al., 2023* used a paradigm similar to Experiment 1 and found that, while resting force biases may initially reflect the velocity at which the arm (passively) approached each test position, this velocity dependence dissipated 2 s after the end of passive movement. In our study, resting biases were measured beginning 2 s *after* the end of movement, which makes a central rather than a reflex mechanism a more likely culprit.

Moreover, it has been shown that joint stiffness is reduced during movement compared to holding control (*Bennett et al., 1992*; *Rack and Westbury, 1974*). Along similar lines, muscle spindle activity – which may modulate stiffness – scales with extrafusal muscle fiber activity (such as muscle exertion involved in holding) and forces acting through the tendon (*Blum et al., 2020*). Such observations could, in principle, explain why we were unable to detect a relationship between resting biases and active movement control, but we readily found a relationship between resting biases and active

holding control: reduced joint stiffness during movement could scale down the influence of resting abnormalities. There are two issues with this explanation, however. First, it is debatable whether this should be considered an alternative explanation per se: stiffness modulation could be, in total or in part, the manifestation of a central movement/posture CST/RST mechanism similar to the one we propose in our conceptual model. For example, *Blum et al., 2020* argue that muscle spindle firing depends on both peripheral and central factors. Second, increased stiffness would not necessarily help detect differences in how active postural control responds to within-resting-posture vs. out-of-resting-posture perturbations. This is because an overall increase in stiffness would likely increase resistance to perturbations in any direction. Interestingly, *Bennett et al., 1992* found that, while stiffness was modulated with elbow angle and gravity during movement, this effect was absent during posture maintenance. This suggests that both the patterns of resting biases we observe in Experiment 1 (tendency towards elbow flexion) and their increase under gravity (i.e. without arm support) cannot be explained by differences in stiffness.

An inherent limitation of our finding that resting biases had no clear effect on active movement is that, being a negative finding, it does not prove there is an absence of effect. Nevertheless, our results suggest that any influence of resting biases on active movement may be subtle or suppressed under provided arm support. This contrasts with the clear connection between resting biases and active holding control that we also found. Moreover, additional analysis demonstrated that the relative asymmetry in how the motor system responds to perturbations that are against vs. in line with resting biases was greater in the holding compared to the movement case (*Figure 8*). Another limitation in this comparison is that the perturbations we used to probe active movement vs. active postural control were different from each other, with each perturbation tailored to each modality. Still, our conceptual model does not reject the idea that resting abnormalities may spill into active movement under the right conditions. The exact relationship between these two modalities may be elucidated in further work, potentially by systematically titrating the amount of arm support to find the point at which resting biases begin to substantially affect active movement.

**Table 1.** Patient characteristics.
FM-UE: Fugl-Meyer Assessment for the Upper Extremity; ARAT: Action Research Arm Test.

| ID | Age (5 years range) | Sex | Time since stroke | Handed-ness | Paretic arm | FM-UE (/66) | ARAT (/57) |
|---|---|---|---|---|---|---|---|
| S001 | 76–80 | M | 2 years | Right | Left | 57.5 | 57 |
| S002 | 51–55 | M | 6 years | Right | Left | 40 | 47.5 |
| S003 | 66–70 | F | 7 years | Right | Right | 34.5 | 19 |
| S004 | 26–30 | F | 5 years | Right | Left | 55.5 | 43.5 |
| S005 | 76–80 | M | 13 months | Right | Right | 43.5 | 34 |
| S007 | 51–55 | F | 2 months | Left | Right | 63 | 57 |
| S008 | 51–55 | F | 14 months | Right | Left | 41 | 25 |
| S009 | 56–60 | F | 5 years | Right | Left | 22 | 3 |
| S010 | 66–70 | M | 5 years | Right | Left | 20 | 12 |
| S011 | 41–45 | F | 20 months | Right | Right | 64 | 57 |
| S012 | 46–50 | M | 6 years | Right | Left | 18.5 | 6.5 |
| S013 | 66–70 | M | 9 years | Right | Left | 14 | 8 |
| S014 | 41–45 | F | 16 months | Right | Left | 40 | 39.5 |
| S015 | 61–65 | F | 10 years | Right | Left | 22 | 4.5 |
| S016 | 36–40 | F | 21 months | Amb. | Right | 62.5 | 57 |
| S017 | 46–50 | M | 3 months | Right | Left | 15 | 3 |

**Table 2.** Summary of patient and control characteristics.
FM-UE: Fugl-Meyer Assessment for the Upper Extremity (/66); ARAT: Action Research Arm Test (/57). MoCA: Montreal Cognitive Assessment (/30). Here, ± indicates standard deviation.

|  | Stroke patients | Controls |
| --- | --- | --- |
| N | 16 | 9 |
| Age | 58.5±17.8 | 62.6±15.2 |
| Gender | 7 M/9 F | 3 M/6 F |
| Paretic side | 11 L/5 R | n/a |
| FM-UE | 38.3±18.2 | 66.0±0.0 |
| ARAT | 29.6±21.8 | 57.0±0.0 |
| MoCA | 24.9±3.1 | 28.1±1.6 |
| Time since stroke | [2 months,10 years] | n/a |

## Conclusions

Our examination of the interplay between abnormalities in moving and holding in stroke patients making planar reaching movements suggests the existence of two functional modes of control, likely constructed out of differing configurations of the CST and the RST. The components of the paretic syndrome – loss of dexterity, weakness, abnormal resting posture, and intrusive synergies – provide insight into how a normal movement is assembled by descending systems. To the degree that hemiparesis is a consequence of the CST losing, to varying degrees, both its direct control over motor neurons and its modulatory influence on the RST, then physiological and behavioral interventions that augment the residual CST may have a plurality of benefits. In support of this, in a recent study of epidural stimulation of the cervical spinal cord in two chronic stroke patients (the rationale behind the stimulation being to amplify residual CST commands), multiple hemiparetic components – strength, dexterity, synergy abnormalities – improved simultaneously (*Powell et al., 2023*).

## Materials and methods
### Participants and ethics statement

Sixteen stroke patients (age: 58.5±17.8 [mean ± standard deviation], nine female) and nine age-range matched healthy control participants (age: 62.6±15.2, six female) were recruited for this study. Sample size was based on similar studies using kinematic and kinetic assessments in patients with stroke (*Beer et al., 2007*; *Beer et al., 2004*; *Bourbonnais et al., 1989*; *Levin, 1996*; *Levin et al., 2000*; *Mani et al., 2013*; *McPherson et al., 2018*; *Sangani et al., 2007*; *Schaefer et al., 2009*; *Sukal et al., 2007*; *Zackowski et al., 2004*). *Table 1* shows details for each patient, whereas *Table 2* shows summary demographics and assessment metrics for patients and controls. Procedures were approved by the Johns Hopkins Institutional Review Board (Protocol # NA_00037510), and participants provided written informed consent in accordance with the Declaration of Helsinki.

### Eligibility criteria

We recruited patients with hemiparesis due to stroke. To be eligible for the study, patients had to be adults, exhibit some movement with the affected arm, and be able to provide informed consent and understand the tasks involved. Exclusion criteria were marked cognitive impairment (assessed based on the Montreal Cognitive Assessment, MoCA, cutoff of 20); severe aphasia or ideomotor apraxia, neglect, or hemianopia; and orthopedic or pain issues.

### Task details

Participants were seated in a chair and grasped the handle of a robotic arm; the handle moved either passively (by itself, Experiment 1) or actively (by the participant, Experiment 2) on the horizontal plane. In Experiment 2 and in two out of four conditions of Experiment 1, participants' lower arm was supported using a custom-made air-sled (*Figure 1C*). Above the plane was a screen that blocked

direct vision of the arm; on this opaque screen, we continuously projected a cursor indicating hand position (diameter: 3 mm), as well as the currently active target (diameter: 10 mm). Handle position was recorded at 200 Hz, whereas subject-produced forces on the handle were recorded using a 6-axis force transducer. Experiments typically began with the paretic arm (see specific details below); for healthy controls, an arm was randomly assigned as primary, with its schedule matching that of the paretic arm in stroke patients (this arm was used for comparisons with patients' paretic data). We decided to prioritize the paretic arm, since the primary analyses in this paper involve comparing different metrics taken from the paretic arm. We note that this absence of paretic/non-paretic counterbalancing could potentially have a limiting effect on interpreting paretic vs. non-paretic comparisons; these comparisons, however, are secondary, and we see no reason why any ordering effects could have more than a minimal effect on them.

## Experiment 1: Measuring resting postural abnormalities

Following a previous paradigm (*Kanade-Mehta et al., 2023*; *Laczko et al., 2017*; *Simo et al., 2013*), Experiment 1 assessed resting postural forces by having the robot passively move participants to different positions in a 2D workspace and hold them still in each position while it measured the forces they inadvertently exerted. Participants were instructed to maintain grasp on the robotic handle but otherwise rest and not resist the robot's motion as it slowly (5 s movement time) moved from one position to the next and held them still (for an additional 5 s). The array of positions (see *Figure 1*) could vary from one participant to the next, with each position visited three times for each block. During the passive moving and holding, a 3-mm white cursor indicating handle position and a 10-mm yellow disk indicating the destination of the passive movement were displayed. Although not essential from the participant's point of view, this allowed the experimenter to monitor the status of the experiment.

Each participant completed four blocks, two with each arm and in each arm support condition (air sled, no air sled); the typical order was {paretic, no support} -> {non-paretic, no support} -> {paretic, air sled support} -> {non-paretic, air sled support}; however, in four individuals, we completed the two paretic blocks first as their hand had to be secured to the handle (with self-adherent wrap) for a stable grasp.

Each block typically took less than 10 min to complete, with Experiment 1 lasting about 40 min including breaks.

## Experiment 2: Assessing reaching control

Using the same workspace as Experiment 1, Experiment 2 assessed motor control in a reaching task. Participants made 10 cm point-to-point reaches across an array of five targets (diameter: 10 mm) within the workspace (*Figure 3A*), sampling 8 different movement directions. A white cursor (diameter: 3 mm), indicating hand position, was visible throughout the experiment. Participants were instructed to try and stop at each target within a 600–800ms window after movement onset. At the end of the movement, feedback was provided to indicate whether they were too fast (time <600ms, target turning red), too slow (time >800ms, target turning blue), or within the right time range (target 'exploding' with a chirping sound).

The experiment was divided into blocks of 96 movements each (12 in each of the 8 movement directions). It began with three blocks with the paretic arm, followed by three more with the non-paretic arm, two more with the paretic arm, and ending with two blocks with the non-paretic arm. The first block with each arm was a familiarization block. Arm support (air sled) was provided throughout the experiment, and breaks were given between blocks.

Except for two participants, who performed Experiment 1 and Experiment 2 on different days due to limitations in their schedule, the entirety of each session – consisting of Experiment 1, Experiment 2, and standard assessments - took place on the same day and typically lasted about 3.5 hr with breaks given between the blocks as necessary.

Most trials (two-thirds) consisted of unperturbed movements to assess reaching control. In the remaining third of reaches, a 12 N, 70ms bell-shaped force pulse that was 12 N at its peak was applied by the robot lateral to the ideal movement direction (i.e. the direction formed between the center of the start position and the center of the target) after participants reached 2 cm away from the starting position (*Fine and Thoroughman, 2006*; *Smith and Shadmehr, 2005*). On half of these trials, the

perturbation was oriented leftwards with respect to the movement (counterclockwise pulse) and the other half rightwards (clockwise pulse).

A fraction of trials in each block (20/96) imposed a perturbation after movement in order to assess active holding control. For these trials, the holding time at the target was extended by 5–7 s, during which participants were instructed to hold still on the target (to remind them, the word 'HOLD' was shown close to the target). During this extended hold period, a 6 N force was gradually imposed over 2 s in one of four different directions (45°, 135°, 225°, 315°), held constant for a pseudorandom time interval uniformly ranging from 3 to 5 s, and then abruptly released. We refer to this as 'release perturbation' throughout the paper. Each block presented each position/release perturbation direction combination exactly once.

Thus, in summary, each 96-movement block consisted of 64 unperturbed movements and 32 movements perturbed with a force pulse (16 clockwise and 16 counter-clockwise). For 20 out of the 96 movements in each block, the hold period was extended to test the hold perturbation (four trials for each of the five target locations, each one of the four trials testing one perturbation direction as shown in *Figure 7C*).

A different perturbation was applied to movement (pulse) as compared to holding (release) so as to maximize disruption for each type of control, rather than to enable direct comparison between them. For holding control, instead of an abruptly imposed force similar to the pulse used to probe movement control, we used a perturbation in which the arm first held against a 6 N force which was then abruptly released. The rationale was that this force preloading would further engage active holding control and thus be better suited to examine it. In line with this, recent work that examined static hold under both types of perturbation – one in which a load is abruptly imposed vs. one in which a load is abruptly removed as we did here – found that the latter was more difficult (*Lowrey et al., 2019*) and hence a better assay for unmasking a holding abnormality.

## Data analysis

Analysis was performed using MATLAB (Mathworks, Natick MA). For Experiment 1, we averaged resting postural forces for the last 3 s of the 5 s passive holding period for each trial. We excluded the first 2 s in order to avoid measuring potential increases in muscle tone that may arise due to the velocity by which the robot passively brought the arm to each measurement position (i.e. avoiding potential effects of spasticity). Recent work using a similar task to examine resting force biases found that the effects of the velocity at which the robot brought the arm to a position are present for 2 s after the end of the passive movement, but then dissipate (*Kanade-Mehta et al., 2023*). To obtain a measure of the average resting postural force at each position for each individual and condition, we further averaged forces across the three visits to the same position. For comparisons, we focused on resting postural forces on the five positions shown in *Figure 1D*; these forces were obtained directly (when the exact positions were sampled for the individual) or through interpolation (7/16 patients and 3/9 controls).

For Experiment 2, movement onset was defined as the moment in which participants' velocity from the starting position exceeded 3.5 cm/s, movement end was defined as the moment the participant was within the target and moving at a speed of less than 3.5 cm/s. The initial reaching angle was calculated between hand position at movement onset and 150ms later; endpoint reaching angle was calculated between the position of the hand when it crossed within 2 cm of the target and 150ms later. In pulse trials, settling time was defined as the time taken from pulse onset to the first moment absolute lateral velocity dipped below 2 cm/s and remained so for at least 100ms (or the movement ended). In release perturbation trials, settling time was defined as the time taken from perturbation onset (release of holding force) to the moment when velocity dipped below 2 cm/s (and remained below that amount for at least 100ms) and the distance from the target was less than 2 cm.

## Data exclusion criteria

In Experiment 1, some trials (7.5%) were flagged as erroneous after visual inspection of force and movement profiles. Erroneous here refers to trials where forces appeared unstable and/or there was movement during the robot hold period (please see *Figure 1E* illustrating an example of one such trial, blue curves on the third panel). We excluded these trials from our main analyses. To ensure that this exclusion did not bias our findings, we repeated our analyses including these trials and obtained

similar results (*Figure 5—figure supplement 4*, *Figure 6—figure supplement 4*, *Figure 7—figure supplement 6*).

In Experiment 2, we excluded as outliers movements in which initial movement direction (150ms after movement onset) was ≥90° away from target direction. This excluded 0.95% of patients' movements and 0.33% of controls' movements.

Moreover, in some release perturbation trials, patients took a long time to reach the stabilization criterion described in the previous section; mistakenly, our setup limited its recording time to only the first 2 s after force release. The exact time to stabilization thus could not be measured for these particular trials, so they had to be excluded from analysis. Though only 13.2 ± 3.3% (mean ± SEM) of paretic stabilization trials were thus excluded in the patient population (1.4 ± 0.4% in their non-paretic side, 2.3±2.0% in controls), there were three patients for which excluded trials were 25% or more of all paretic trials. To ensure there are no systematic effects of this issue, we repeated the analysis of *Figure 7F* (a) by excluding these three patients altogether or (b) by assigning a value of 2.0 s to the affected trials. In both cases, we obtained results similar to our main analysis (*Figure 7—figure supplement 5*).

## Stability of resting posture bias measurements in Experiment 1

*Figure 1E* shows a few examples of the within-trial evolution of force bias measurements during the 5 s hold period, illustrating the high within-trial temporal stability in our dataset – particularly after the 2 s mark, which begins the measurement window. To systematically assess the stability of our estimates of resting force biases *across* different trials, we estimated the variance attributed to different measurements at the same location (in contrast to the variance attributed to different locations, conditions, and patients). We found that this measurement variance was only 9.0% of the total variance for resting bias magnitude.

## Statistical comparisons

In Experiment 1, we used an ANOVA to investigate any effect of conditions {Position (distant targets [furthest three positions] vs. near targets [closest two positions]), Support (with/without air sled), FM-UE (continuous)} and their interactions.

In Experiment 2, we used paired t-tests for the within-subject comparisons of outcome variables (For unperturbed reaching: initial reaching angle, endpoint reaching angle; for in-movement pulse perturbations: maximum deviation in the pulse direction, settling time; for holding release perturbations: maximum deviation in the perturbation direction; time to stabilization; path to stabilization) against the corresponding resting biases measured in Experiment 1. We performed two types of comparisons:

First, our main analyses took, for each individual, the two instances for which the resting biases would have the strongest opposing effects. For variables related to unperturbed reaching, we would take the movement direction for which the resting biases would be oriented the strongest CCW vs. the movement direction for which the resting biases would be oriented the strongest CW (two out of eight directions). The corresponding trials (32 for each of the two instances per participant) were averaged for each individual. For variables related to responses to pulse perturbation, we would take the two movement directions for which the resting bias at start would most oppose vs. most align with the pulse. The corresponding trials (eight for each of the two instances) were averaged for each individual. For variables related to responses to the release perturbation, we took the two perturbation direction/hold position combinations for which the resting bias on that position would most oppose vs. most align with the perturbation. The corresponding trials (four for each of the two instances) were averaged for each individual.

Second, we used linear regression to calculate, for each individual, the sensitivities of outcome variables from Experiment 2 to the corresponding resting biases measured in Experiment 1 using data from all trials (*Figure 5—figure supplement 2*, *Figure 6—figure supplement 2*, and *Figure 7—figure supplement 2*), rather than the most extreme instances. We then tested whether the average of these per-individual sensitivities was significantly greater than zero using a paired t-test.

We used the circular statistics toolbox CircStat (*Berens, 2009*) to estimate the circular mean ± SEM for the direction of resting biases shown in *Figure 3B*, right.

## Fugl-Meyer assessments

Assessments were separately scored by AMH and KK with scores subsequently averaged (hence some scores having decimal values). For cases of substantial score differences (3 points or more), scores were again reviewed by both raters together.

## Acknowledgements

We would like to thank Stuart Baker and Maurice Smith for helpful discussions about this manuscript. Funding Support for this work was provided by the Sheikh Khalifa Stroke Institute to AMH and KK, a T32 Fellowship by the National Institute of Neurological Diseases and Stroke to AMH (T32NS100663), and a Career Development Award by the American Heart Association to AMH (25CDA1439419). Funding sources were not involved in study design, data collection and interpretation, or the decision to submit the work for publication.

## Additional information

### Funding

| Funder | Grant reference number | Author |
| --- | --- | --- |
| Sheikh Khalifa Stroke Institute | | Alkis M Hadjiosif<br>Kahori Kita |
| National Institutes of Health | 5T32NS100663 | Alkis M Hadjiosif |
| American Heart Association | 10.58275/AHA.25CDA1439419.pc.gr.229633 | Alkis M Hadjiosif |

The funders had no role in study design, data collection and interpretation, or the decision to submit the work for publication.

### Author contributions

Alkis M Hadjiosif, Conceptualization, Data curation, Software, Formal analysis, Validation, Investigation, Visualization, Methodology, Writing – original draft, Project administration, Writing – review and editing; Kahori Kita, Data curation, Investigation, Writing – review and editing; Scott T Albert, Software, Writing – review and editing; Robert A Scheidt, Conceptualization, Software, Methodology, Writing – review and editing; Reza Shadmehr, Conceptualization, Resources, Software, Supervision, Funding acquisition, Writing – review and editing; John W Krakauer, Conceptualization, Resources, Supervision, Funding acquisition, Writing – review and editing

### Author ORCIDs

Alkis M Hadjiosif ⓘ https://orcid.org/0000-0001-8823-3631
Kahori Kita ⓘ https://orcid.org/0000-0001-5002-2863
Scott T Albert ⓘ https://orcid.org/0000-0001-9140-1077
Robert A Scheidt ⓘ https://orcid.org/0000-0002-2024-5051
Reza Shadmehr ⓘ https://orcid.org/0000-0002-7686-2569
John W Krakauer ⓘ https://orcid.org/0000-0002-4316-1846

### Ethics

Procedures were approved by the Johns Hopkins Institutional Review Board, and participants provided written informed consent in accordance with the Declaration of Helsinki.

Reviewer #1 (Public review): https://doi.org/10.7554/eLife.90780.4.sa1
Reviewer #2 (Public review): https://doi.org/10.7554/eLife.90780.4.sa2
Author response https://doi.org/10.7554/eLife.90780.4.sa3

## Additional files

**Supplementary files**
MDAR checklist

**Data availability**
Data and analysis code supporting the findings in this paper are maintained at https://osf.io/hufy8/.

The following dataset was generated:

| Author(s) | Year | Dataset title | Dataset URL | Database and Identifier |
|---|---|---|---|---|
| Hadjiosif AM, Kahori K, Albert ST, Scheidt RA, Shadmehr R, Krakauer JW | 2025 | Separating the control of moving and holding in post-stroke arm paresis | https://osf.io/hufy8/ | Open Science Framework, hufy8 |

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
