## [Editor Report · eLife Assessment]

This **important** study extends the previous interesting work of this group to address the potentially different control of movement and posture. Through experiments in which stroke participants used a robotic manipulandum, the authors provide **solid** evidence supporting a lack of a relation between the resting force postural bias they measure (closely related to the flexor synergy in stroke) and kinematic deficits during movement. Based on these results, the authors propose a conceptual framework that differentially weights the two main descending pathways (corticospinal tract and reticulospinal tract) for neurologically intact and stroke patients.

---

## [Referee Report · Reviewer #1 (Public review)]

This study extends the previous interesting work of this group to address the potentially differential control of movement and posture. Their earlier work explored a broad range of data to make the case for a downstream neural integrator hypothesized to convert descending velocity movement commands into postural holding commands. Included in that data were observations from people with hemiparesis due to stroke. The current study uses similar data, but pushes into a different, but closely related direction, suggesting that these data may address the independence of these two fundamental components of motor control. The study makes observations about the different expression movement deficits during postural fixation and movement, and the different effect of force perturbations during these periods, consistent with their hypothesis that movement and postural control are separate motor functions. They speculate that the appearance of the stereotypic flexor synergies characteristic of stroke, are the result of a breakdown of this normal separation between the two control modes.

Comments on revisions:

I had only two very trivial comments in the previous version. One was simply a figure that was mistakenly not updated, and the other was the use of the terms "proximal" and "distal" to describe the location of a target. Both have been corrected.

---

## [Referee Report · Reviewer #2 (Public review)]

The reported findings by Hadjiosif and colleagues address an important question in sensorimotor neuroscience related to the idea that movement and postural control are regulated by unique circuits. To explain the reported compromised postural control for stroke patients, the authors propose a conceptual framework that differentially weights corticospinal tract and reticulospinal tract for neurologically intact and stroke patients. Based on the currently reported findings and experimental design, the interpretation of the authors provides support to this idea.

The authors have done well to include a limitations paragraph in their discussion. While it is difficult to truly compare across many of the experimental conditions to draw any strong conclusions, the authors have included additional analyses and a limitations paragraph highlighting some weaknesses in the paper.

---

## [Author Response]

The following is the authors’ response to the previous reviews

**Public Reviews:**

**Reviewer #1 (Public review):**
This study extends the previous interesting work of this group to address the potentially differential control of movement and posture. Their earlier work explored a broad range of data to make the case for a downstream neural integrator hypothesized to convert descending velocity movement commands into postural holding commands. Included in that data were observations from people with hemiparesis due to stroke. The current study uses similar data, but pushes into a different, but closely related direction, suggesting that these data may address the independence of these two fundamental components of motor control. I find the logic laid out in the second sentence of the abstract ("The paretic arm after stroke is notable for abnormalities both at rest and during movement, thus it provides an opportunity to address the relationships between control of reaching, stopping, and stabilizing") less then compelling, but the study does make some interesting observations. Foremost among them, is the relation between the resting force postural bias and the effect of force perturbations during the target hold periods, but not during movement. While this interesting observation is consistent with the central mechanism the authors suggest, it seems hard to me to rule out other mechanisms, including peripheral ones. These limitations should should be discussed.

Thank you for summarizing our work. Note we have improved the logic in our abstract (…”providing an opportunity to ask whether control of these behaviors is independently affected in stroke”) based on your comments as outlined in our previous revision. We now extensively discuss limitations and potential alternative mechanisms in greater detail, in a dedicated section (lines 846-895; see response to reviewer 2 for further details).

**Reviewer #2 (Public review):**
Summary:Here the authors address the idea that postural and movement control are differentially impacted with stroke. Specifically, they examined whether resting postural forces influenced several metrics of sensorimotor control (e.g., initial reach angle, maximum lateral hand deviation following a perturbation, etc.) during movement or posture. The authors found that resting postural forces influenced control only following the posture perturbation for the paretic arm of stroke patients, but not during movement. They also found that resting postural forces were greater when the arm was unsupported, which correlated with abnormal synergies (as assessed by the Fugl-Meyer). The authors suggest that these findings can be explained by the idea that the neural circuitry associated with posture is relatively more impacted by stroke than the neural circuitry associated with movement. They also propose a conceptual model that differentially weights the reticulospinal tract (RST) and corticospinal tract (CST) to explain greater relative impairments with posture control relative to movement control, due to abnormal synergies, in those with stroke.

Thank you for the brief but comprehensive summary. We would like to clarify one point: we do not suggest that our findings are necessarily due to the neural circuitry associated with posture being more impacted than the neural circuitry associated with movement. (rather, our conceptual model suggests that increased outflow through the (ipsilateral) RST, involved in posture, compensates for CST damage, at the expense of posture abnormalities spilling over into movement). Instead, we suggest that the neural circuitry for posture vs. movement control remains relatively separate in stroke, with impairments in posture control not substantially explaining impairments in movement control.

Comments on revisions:The authors should be commended for being very responsive to comments and providing several further requested analyses, which have improved the paper. However, there is still some outstanding issues that make it difficult to fully support the provided interpretation.

Thank you for appreciating our response to your earlier comments. We address the outstanding issues below.

The authors say within the response, "We would also like to stress that these perturbations were not designed so that responses are directly compared to each other ***(though of course there is an *indirect* comparison in the sense that we show influence of biases in one type of perturbation but not the other)***." They then state in the first paragraph of the discussion that "Remarkably, these resting postural force biases did not seem to have a detectable effect upon any component of active reaching but only emerged during the control of holding still after the movement ended. The results suggest a dissociation between the control of movement and posture." The main issue here is relying on indirect comparisons (i.e., significant in one situation but not the other), instead of relying on direct comparisons. Using well-known example, just because one group / condition might display a significant linear relationship (i.e., slope_1 > 0) and another group / condition does not (slope_2 = 0), does not necessarily mean that the two groups / conditions are statistically different from one another [see Figure 1 in Makin, T. R., & Orban de Xivry, J. J. (2019). Ten common statistical mistakes to watch out for when writing or reviewing a manuscript. eLife, 8, e48175.].

We agree and are well aware of the limitation posed by an indirect comparison – hence the language we used to comment on the data (“did not seem”, “suggest”, etc.). To address this limitation, we performed a more direct comparison of how the two types of perturbations (moving vs. holding) interact with resting biases. For this comparison, we calculated a Response Asymmetry Index (RAI):\begin{document}$$\displaystyle R A I=\frac{r_{A}-r_{O}}{r_{A}+r_{O}}$$\end{document}

Above, 𝑟_𝐴_ is the response on direction where resting bias is most-aligned with the perturbation, and 𝑟_𝑂_ is the response on direction where resting bias is most-opposed to the perturbation.

We calculated RAIs for two response metrics used for both moving and holding perturbations: maximum deviation and time to stabilization/settling time. For these two response metrics, positive RAIs indicate an asymmetry in line with an effect of resting bias.

The idea behind the RAI is that, while the magnitude of responses may well differ between the two types of perturbations, this will be accounted for by the ratio used to calculate the asymmetry. The same approach has been used to assess symmetry/laterality across a variety of different modalities, such as gait asymmetry (Robinson et al., 1987), the relative fMRI activity in the contralateral vs. ipsilateral sensorimotor cortex while performing a motor task (Cramer et al., 1997), or the relative strength of ipsilateral vs. contralateral responses to transcranial magnetic stimulation (McPherson et al., 2018). Notably, the normalization also addresses potential differences in overall stiffness between holding vs. moving perturbations, which would similarly affect aligned and opposing cases (see our response to your following point).

Figure 8 shows RAIs we obtained for holding (red) vs. moving/pulse (blue) perturbations. For the maximum deviation (left), there is more asymmetry for the holding case though the pvalue is marginal (p=0.088) likely due to the large variability in the pulse case (individual values shown in black dots). For time to stabilization/settling time (right) the difference is significant (p=0.0048). Together, these analyses indicate that resting biases interact substantially more with holding compared to movement control, in line with a relative independence between these two control modalities. We now include this panel as Figure 8, and describe it in Results (lines 587-611).

Note that even a direct comparison does not prove that resting biases and active movement control are perfectly independent. We now discuss these issues in more depth, in the new Limitations section suggested by the Reviewer (lines 836-849).

The authors have provided reasonable rationale of why they chose certain perturbation waveforms for different. Yet it still holds that these different waveforms would likely yield very different muscular responses making it difficult to interpret the results and this remains a limitation. From the paper it is unknown how these different perturbations would differentially influence a variety of classic neuromuscular responses, including short-range stiffness and stretch reflexes, which would be at play here.Much of the results can be interpreted when one considers classic neuromuscular physiology. In Experiment 1, differences in resting postural bias in supported versus unsupported conditions can readily be explained since there is greater muscle activity in the unsupported condition that leads to greater muscle stiffness to resist mechanical perturbations (Rack, P. M., & Westbury, D. R. (1974). The short-range stiffness of active mammalian muscle and its effect on mechanical properties. The Journal of physiology, 240(2), 331-350.). Likewise muscle stiffness would scale with changes in muscle contraction with synergies. Importantly for experiment 2, muscle stiffness is reduced during movement (Rack and Westbury, 1974) which may explain why resting postural biases do not seem to be impacting movement. Likewise, muscle spindle activity is shown to scale with extrafusal muscle fiber activity and forces acting through the tendon (Blum, K. P., Campbell, K. S., Horslen, B. C., Nardelli, P., Housley, S. N., Cope, T. C., & Ting, L. H. (2020). Diverse and complex muscle spindle afferent firing properties emerge from multiscale muscle mechanics. eLife, 9, e55177.). The concern here is that the authors have not sufficiently considered muscle neurophysiology, how that might relate to their findings, and how that might impact their interpretation. Given the differences in perturbations and muscle states at different phases, the concern is that it is not possible to disentangle whether the results are due to classic neurophysiology, the hypothesis they propose, or both. Can the authors please comment.

It is possible that neuromuscular physiology may explain part of our results. However, this would not contradict our conceptual model.

Regarding Experiment 1, it is possible that stiffness would scale with changes in background muscle contraction as the reviewer suggests. Indeed, Bennett and al.(Bennett et al., 1992) used brief perturbations on the wrist to assess elbow stiffness, finding that, during movement, stiffness was increased in positions with a higher gravity load (and, in general, in positions where the net muscle torque was higher). However, during posture maintenance (like in our Experiment 1), they found that stiffness did not vary with (elbow) position or gravity load (two characteristics of our findings in Experiment 1):

“The observed stiffness variation was not simply due to passive tissue or other joint angle dependent properties, as stiffnesses measured during posture were position invariant. Note that the minimum stiffness found in posture was higher than the peak stiffness measured during movement, and did not change much with the gravity load.” (illustrated in Fig. 5 of that paper)

We thus find it very unlikely that stiffness explains the difference between the supported vs. unsupported conditions in Experiment 1.

Even if stiffness modulation between the supported vs. unsupported conditions could explain our finding of stronger posture biases in the latter case, it would not be incompatible with our interpretation of increased RST drive: increased stiffness would potentially magnify the effects of the RST drive we propose to drive these resting biases. It is possible that the increase in resting biases under conditions of increased muscle contraction (lack of arm support) is mediated through an increase in muscle stiffness. In other words, the increase in resting biases may not directly reflect additional RST outflow per se, but the scaling, through stiffness, of the same magnitude of RST outflow. Understanding this interaction was beyond the scope of our experiment design; in line with this, we briefly comment about it in our Limitations section.

Regarding Experiment 2, stiffness has indeed been shown to be lower during movement, and we now comment the potential effect of this on our results in the “Limitations” section (lines 815-830, replicated below). Importantly, for the case of holding perturbations, the increased stiffness associated with holding would increase resistance to both extension and flexion-inducing perturbations. Thus, higher stiffness would be unlikely to explain our finding whereby resting biases resist or aggravate the effects of holding perturbations depending on perturbation direction. In addition, the framework in Blum et al., that describes how interactions between alpha and gramma drive can explain muscle activity patterns, does not rule out central neural control of stiffness: “muscle spindles have a unique muscle-within-muscle design such that their firing depends critically on both peripheral and central factors” (emphasis ours). It may be, for example, that gamma motoneurons controlling muscle spindles and stiffness are modulated from input from the reticular formation, making this a mechanism in line with our conceptual model.

“Moreover, it has been shown that joint stiffness is reduced during movement compared to holding control (Rack and Westbury, 1974; Bennett et al., 1992). Along similar lines, muscle spindle activity – which may modulate stiffness – scales with extrafusal muscle fiber activity (such as muscle exertion involved in holding) and forces acting through the tendon (Blum et al., 2020). Such observations could, in principle, explain why we were unable to detect a relationship between resting biases and active movement control but we readily found a relationship between resting biases and active holding control: reduced joint stiffness during movement could scale down the influence of resting abnormalities. There are two issues with this explanation, however. First, it is debatable whether this should be considered an alternative explanation per se: stiffness modulation could be, in total or in part, the manifestation of a central movement/posture CST/RST mechanism similar to the one we propose in our conceptual model. For example, (Blum et al., 2020) argue that muscle spindle firing depends on both peripheral and central factors. Second, increased stiffness would not necessarily help detect differences in how active postural control responds to within-resting-posture vs. out-of-resting-posture perturbations. This is because an overall increase in stiffness would likely increase resistance to perturbations in any direction.”

The authors should provide a limitations paragraph. They should address (1) how they used different perturbation force profiles, (2) the muscles were in different states which would change neuromuscular responses between trial phase / condition, (3) discuss a lack of direct statistical comparisons that support their hypothesis, and (4) provide a couple of paragraphs on classic neurophysiology, such as muscle stiffness and stretch reflexes, and how these various factors could influence the findings (i.e., whether they can disentangle whether the reported results are due to classic neurophysiology, the hypothesis they propose, or both).

Thank you for your suggestion. We now discuss these points in a separate paragraph (lines 846895), bringing together our previous discussion on stretch reflexes, our description of different perturbation types, and the additional issues raised by the reviewer above.

**Recommendations for the authors:**

**Reviewer #1 (Recommendations for the authors):**
The authors have responded well to all my concerns, save two minor points.Figure 2 appears to be unchanged, although they describe appropriate changes in the response letter.

Thank you for catching this error – we now include the updated figure (further updated to use the terms near/distant in place of proximal/distal).

I still take issue with the use of proximal and distal to describe the locations of targets. Taking definitions somewhat randomly from the internet, "The terms proximal and distal are used in structures that are considered to have a beginning and an end," and "Proximal and distal are anatomical terms used to describe the position of a body part in relation to another part or its origin." In any case, the hand does not become proximal just because you bring it to your chest. Why not simply stick to the common and clearly defined terms "near" and "distant"?

Point taken. We have updated the paper to use the terms near/distant.

Additional changes/corrections not outlined above

We now include a link to the data and code supporting our findings (https://osf.io/hufy8/). In addition, we made several minor edits throughout the text to improve readability, and corrected occasional mislabeling of CCW and CW pulse data. Note that this correction did not alter the (lack of) relationship between resting biases and responses to perturbations during active movement.

Response letter references

Bennett D, Hollerbach J, Xu Y, Hunter I (1992) Time-varying stiffness of human elbow joint during cyclic voluntary movement. Exp Brain Res 88:433–442.

Blum KP, Campbell KS, Horslen BC, Nardelli P, Housley SN, Cope TC, Ting LH (2020) Diverse and complex muscle spindle afferent firing properties emerge from multiscale muscle mechanics. Elife 9:e55177.

Cramer SC, Nelles G, Benson RR, Kaplan JD, Parker RA, Kwong KK, Kennedy DN, Finklestein SP, Rosen BR (1997) A functional MRI study of subjects recovered from hemiparetic stroke. Stroke 28:2518–2527.

McPherson JG, Chen A, Ellis MD, Yao J, Heckman C, Dewald JP (2018) Progressive recruitment of contralesional cortico-reticulospinal pathways drives motor impairment post stroke. J Physiol 596:1211–1225 Available at: https://doi.org/10.1113/JP274968.

Rack PM, Westbury D (1974) The short range stiffness of active mammalian muscle and its effect on mechanical properties. J Physiol 240:331–350.

Robinson R, Herzog W, Nigg BM (1987) Use of force platform variables to quantify the effects of chiropractic manipulation on gait symmetry. J Manipulative Physiol Ther 10:172–176.

Williams PE, Goldspink G (1973) The effect of immobilization on the longitudinal growth of striated muscle fibres. J Anat 116:45.